# Interaction network of human early embryonic transcription factors

Lisa Gawriyski [1,2,3], Zenglai Tan [4,9], Xiaonan Liu [1,9], Iftekhar Chowdhury [1,9], Dicle Malaymar Pinar [1,5,9], Qin Zhang[6], Jere Weltner[2,3], Eeva-Mari Jouhilahti[2,3], Gong-Hong Wei [4,6], Juha Kere [2,3,7] & Markku Varjosalo [1,8]

## Abstract

**Embryonic genome activation (EGA) occurs during preimplantation development and is characterized by the initiation of de novo transcription from the embryonic genome. Despite its importance, the regulation of EGA and the transcription factors involved in this process are poorly understood. Paired-like homeobox (PRDL) family proteins are implicated as potential transcriptional regulators of EGA, yet the PRDL-mediated gene regulatory networks remain uncharacterized. To investigate the function of PRDL proteins, we are identifying the molecular interactions and the functions of a subset family of the Eutherian Totipotent Cell Homeobox (ETCHbox) proteins, seven PRDL family proteins and six other transcription factors (TFs), all suggested to participate in transcriptional regulation during preimplantation. Using mass spectrometry-based interactomics methods, AP-MS and proximity-dependent biotin labeling, and chromatin immunoprecipitation sequencing we derive the comprehensive regulatory networks of these preimplantation TFs. By these interactomics tools we identify more than a thousand high-confidence interactions for the 21 studied bait proteins with more than 300 interacting proteins. We also establish that TPRX2, currently assigned as pseudogene, is a transcriptional activator.**

**Keywords** BioID; ChIP-Seq; EGA; PRDL; ETCHbox
**Subject Categories** Chromatin, Transcription & Genomics; Development

## Introduction

In human embryos, the initiation of de novo transcription (embryonic genome activation, EGA) occurs in waves during the 2- to 8-cell stages (Braude et al, 1988; Vassena et al, 2011; Yan et al, 2013; Xue et al, 2013; Töhönen et al, 2015). The EGA regulation, the transcription factors (TFs) involved in this, and especially their roles in participating and initiating stages of EGA are still poorly defined. Transcriptome sequencing studies on preimplantation embryos have identified differential expression of genes during the EGA-stages, including the downregulation of thousands of transcripts specific to oocytes (Zhang et al, 2009; Vassena et al, 2011; Yan et al, 2013; Piras et al, 2014; Töhönen et al, 2015). A transcription start site-directed study of EGA identified several previously poorly annotated TFs of the paired-like (PRDL) homeobox gene family, including the Eutherian Totipotent Cell homeobox (ETCHbox) family, as potential key regulators of early EGA (Töhönen et al 2015).

Within the ETCHbox family, the EGA-associated genes *LEUTX*, *ARGFX*, *DPRX*, *TPRX1*, and *TPRX2* are thought to have arisen by tandem gene duplication and subsequent asymmetric sequence evolution from homeobox gene *CRX* (*OTX3*) of the Otx gene family (Holland et al, 2007; Maeso et al, 2016). The gene sequences, including the homeodomains, have since undergone extensive divergence, leading to likely changes in gene function (Maeso et al, 2016). We have shown overlapping expression profiling of ETCHbox genes during EGA and suggested a role for LEUTX as a transcriptional activator, the activity of which is reduced by DPRX most likely through direct competition for the DNA binding motif (Töhönen et al, 2015; Jouhilahti et al, 2016; Madissoon et al, 2016). Furthermore, previous studies show genes regulated by ETCHbox-genes overlap (Maeso et al, 2016) and *ARGFX* and *DPRX* were found to be upregulated by TPRX1 (Zou et al, 2022).

In contrast, little is known about the protein interaction networks or function of these proteins, in part because this family is not expressed in somatic tissues but only in early embryonic cells and some cancers. ETCHbox genes are a subset of paired-like homeobox (PRDL) family that includes the *CPHX1*, *CPHX2*, *DUXA*, and *DUXB* genes that also have been shown to be active during preimplantation development (Töhönen et al, 2015; Zou et al, 2022). *DUXA* and *DUXB* are also found to be upregulated by TPRX1 (Zou et al, 2022).

[1]University of Helsinki, Institute of Biotechnology, Helsinki, Finland. [2]Stem Cells and Metabolism Research Program, University of Helsinki, Helsinki, Finland. [3]Folkhälsan Research Center, Helsinki, Finland. [4]Disease Networks Research Unit, Faculty of Biochemistry and Molecular Medicine & Biocenter Oulu, University of Oulu, Oulu, Finland. [5]Department of Molecular Biology and Genetics, Istanbul Technical University, Istanbul, Turkey. [6]Ministry of Education Key Laboratory of Metabolism and Molecular Medicine & Department of Biochemistry and Molecular Biology, School of Basic Medical Sciences, Cancer Institute, Fudan University Shanghai Cancer Center; Department of Oncology, Shanghai Medical College of Fudan University, Shanghai, China. [7]Karolinska Institutet, Department of Biosciences and Nutrition, Huddinge, Sweden. [8]iCAN Digital Precision Cancer Medicine Flagship, University of Helsinki, Helsinki, Finland. [9]These authors contributed equally: Zenglai Tan, Xiaonan Liu, Iftekhar Chowdhury, Dicle Malaymar Pinar. ✉E-mail: markku.varjosalo@helsinki.fi

Recently, methods to produce 8-cell stage-like cells (8CLC) from pluripotent cells have been developed, and *TPRX1* has been proposed to be an 8-cell stage-like marker gene (Mazid et al, 2022; Taubenschmid-Stowers et al, 2022). However, TPRX1 function remains unclear, mainly due to unknown protein interaction and gene regulatory networks of the ETCHbox genes. *ARGFX, LEUTX, ZSCAN4, DUXA*, and *DUXB* have been identified as additional markers of the 8CLC-like transcriptional state (Mazid et al, 2022; Taubenschmid-Stowers et al, 2022; Yoshihara et al, 2022). Recent research suggested that combined *TPRX1, TPRX2,* and *TPRXL* gene knockdowns lead to developmental defects and delays in EGA while *LEUTX* knockdown alone only had minor effects (Zou et al, 2022).

In this study, we characterized the functions of the ETCHbox family members, through their protein-protein interactions (PPIs) and DNA binding activity. We used two complementary mass spectrometry-based interactomics methods, affinity purification (AP) and proximity labeling (BioID), and chromatin immunoprecipitation sequencing (ChIP-Seq) in the Flp-In™ T-REx™ 293 cell lines to derive regulatory networks of these preimplantation regulating transcription factors (TFs). We also analyzed the PRDL factors *OTX1, OTX2, CPHX1, CPHX2, DUXA, DUXB, GSC, PITX1*, and *PITX2*, all of which have been implicated in preimplantation development (Töhönen et al, 2015; Liu et al, 2019; Xia et al, 2019). The PRDL binding site was shown to be enriched in the promoters of 4- and 8-cell stage EGA gene promoters (Töhönen et al, 2015) and in 8-cell stage enhancer regions (Xia et al, 2019). Our protein of interest (bait) set also included *DPPA3*, a direct transcriptional regulation target of LEUTX (Gawriyski et al, 2023); the preimplantation TFs *EGR2, ZNF263*, and *KLF3* (Leng et al, 2019); and *ZSCAN4* (Töhönen et al, 2015; Hendrickson et al, 2017). *PITX2* and *GSC* are lineage-specific markers for meso- and endoderm, and mesendoderm, respectively, and are activated in 8CLCs (Mazid et al, 2022).

Study of the PPIs of these factors reveals a wider network of key proteins potentially involved in preimplantation development as well as key genomic regions where the protein function. We found that most of the PRDL proteins are likely to be transcriptional activators, consistent with their putative transcriptional activation domains, their interactions with coactivators, and their binding of sites proximal to developmentally important genes. Further, we found a large overlap of proximal interactors for all 21 target proteins. These shared proteins represent candidates for further research, including stem cell reprogramming approaches.

# Results

## PRDL proteins contain putative transactivation domains

To establish the identity and functions of proteins involved in preimplantation development, we focused on the ETCHbox gene family (Figs. 1A and EV1A,B). We performed multiple sequence analyses using MAFFT and domain prediction using InterPro and PredictProtein to identify secondary structure and sequence conservation in the ETCHbox family proteins (Figs. 1B and EV1C). For all analyzed proteins, InterPro domain prediction identified a highly conserved full-length N-terminal homeodomain and several disordered regions, but no other known functional domains or motifs (Figs. 1B and EV1D). In most ETCHbox proteins, an N-terminal K50-class homeodomain forms the DNA

binding domain, with potential to bind to a TAATCC or TAAGCT binding motif (Baird-Titus et al, 2006) that is found to be enriched in promoters and enhancers of EGA genes (Töhönen et al, 2015; Katayama et al, 2018). ARGFX has a Q50-class homeodomain, with the potential to bind to a TAATTA, TAATTG, or TAATGG binding motif (Baird-Titus et al, 2006). The protein sequences primarily align through the conserved homeodomain. Other segments of the proteins do not show high sequence similarity, except for TPRX1 and TPRX2, which share high sequence similarity regions also outside the homeodomain region. Through PredictProtein analysis, additional RNA-binding function is predicted for CRX, LEUTX, TPRX1, and TPRX2. All factors are predicted to interact with DNA via their homeodomain.

We have previously shown that LEUTX has a C-terminal transactivation domain (TAD; Katayama et al, 2018). To detect putative TADs in other PRDL family members, we employed an in silico 9aaTAD Prediction Tool and used first a moderately stringent search pattern and a less stringent pattern if there were no moderate stringency hits to look for potential 9aaTAD motifs (Piskacek et al, 2016). TADs are TF scaffold domains which are often involved in transcriptional regulation and coactivation and interact with kinase-inducible domains (Piskacek et al, 2007; Yadav et al, 2017). We found a 100% 9aaTAD motif match in ARGFX with moderately stringent search settings. ARGFX also has conserved regions in its C-terminal end, suggesting further potential functional domains may be present (Fig. 1B). TPRX1, TPRX2, OTX1, OTX2, CRX, CPHX1, CPHX2, KLF3, PITX1, PITX2, GSC, DUXA, and ZNF263 all had a putative 9aaTAD motif match using a low stringency 9aaTAD motif search (Dataset EV1). Currently, 41 human proteins in UniProtKB have a putative 9aaTAD, including the majority of the KLF-family (Piskacek et al, 2021). 9aaTADs have previously been reported in all Yamanaka factors (Piskacek et al, 2021) and many other TFs (Piskacek et al, 2020). In LEUTX, the C-terminal 9aaTAD has been shown to facilitate stable interaction with EP300 and CBP (Gawriyski et al, 2023). Similarly, the tumor protein p53 has a TAD that facilitates its interaction with CBP and EP300, and directly affects its transcriptional regulatory potential (Teufel et al, 2007; Feng et al, 2009).

## Identifying high-confidence interactors for nuclear proteins

To characterize preimplantation TF protein-protein interactions (PPIs), we used two complementary interactomics methods (Fig. EV1B) using the MAC-tag in HEK293 cells. Affinity purification-mass spectrometry (AP-MS) was used to study stable PPIs, and proximity-dependent biotin identification MS (BioID-MS) was used to study dynamic interactions with an protocol optimized for nuclear proteins (Liu et al, 2018; Liu et al, 2020). Flp-In™ T-REx™ 293 cells were chosen as the model system as it represent the "gold standard" in interaction proteomics and expresses a very wide range of different genes. In addition, the transgene expression in these cells is inducible, a feature which is highly needed for studying key signaling molecules such as the transcription factors of which expression can be toxic (Vuoristo et al, 2022) or transform the cells when expressed constitutively. In addition, we and others have optimized the use of this system for studying TFs (Göös et al, 2022; Alerasool et al, 2022). Nuclear

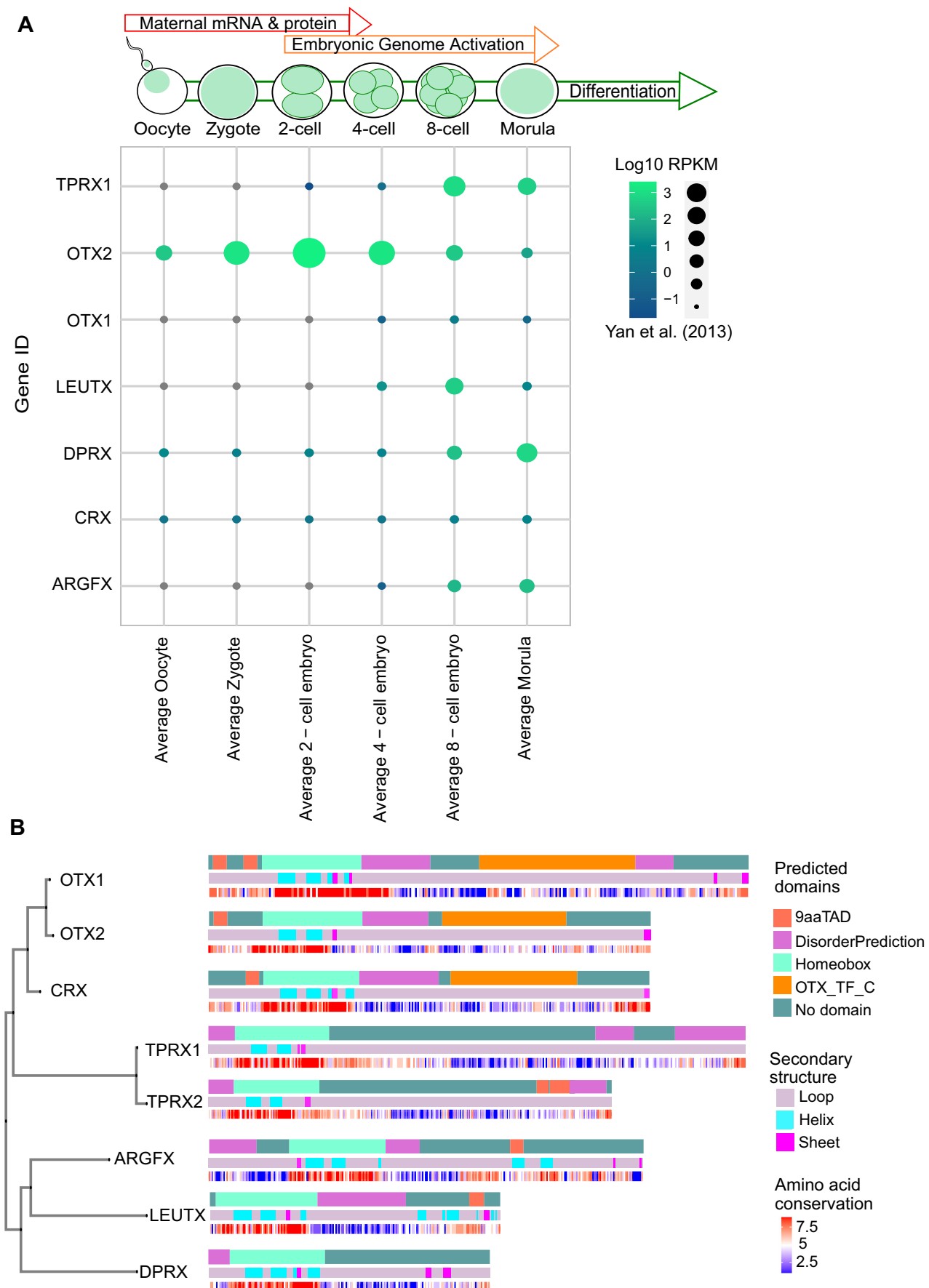

**Figure 1. Expression and phylogenetic analysis of the ETCHbox and OTX gene family during preimplantation development.**

(A) Expression pattern of the ETCHbox and OTX gene family during preimplantation development from the oocyte to the morula stages. Heatmap color and dot size depict log10 RPKM values, as reported in embryonic transcriptomic dataset from Yan et al (2013). Translation occurs from stored maternal mRNA during early stages of preimplantation development, followed by embryonic genome activation (EGA) in later stages. (B) Phylogenetic tree showing the protein sequences and domain predictions for the ETCHbox family and CRX, OTX1, and OTX2 from the Otx gene family. The first row shows protein sequence analysis, with the complete N-terminal homeobox domain shown in light green. The conserved C-terminal region in Otx family members is lost in the ETCHbox family. In most proteins, no other known domains were found, and disordered regions are predicted (shown in light purple). In addition, we performed in silico prediction for 9aaTADs in the ETCHbox proteins (shown in light red). The second row shows the predicted secondary structure, with helixes in cyan and sheets in magenta. The bottom row shows amino acid conservation as a heatmap between red and blue, with red indicating high conservation.

localization signal (NLS) tagged GFP served as a negative control in both approaches. BioID-MS also captures weak and transient interactions, and has high sensitivity for identifying TF interactors (Göös et al, 2022). The MAC-tag combines a HA- and Strep-tag II, as well as abiotin ligase (BirA), enabling affinity purification of bait and interactors using streptavidin/Strep-Tactin (Liu et al, 2018; Liu et al, 2020).

To ensure that our interactomics analyses were as stringent as possible, we constructed a BioID-MS "contaminant" database, similar to the Contaminant Repository for Affinity Purification (CRAPome) (Mellacheruvu et al, 2013), thereby enabling efficient filtering of non-specific interactors. The CRAPome was principally constructed for filtering AP-MS data and cannot efficiently filter for BioID-MS data. In addition, the majority of the CRAPome controls are not nuclear-localized. Therefore, in our targeted BioID-MS contaminant dataset, we included 113 GFP control purifications (64 without a nuclear localization signal, 23 with a myristoylation signal, 2 with a nuclear export signal, and 25 with a nuclear localization signal) (Appendix Fig. S1A,B). In total, our GFP BioID-MS contaminant database contains 4014 identified probable unspecific interactors (Dataset EV2).

## Protein interactome analysis: TPRX2 is an activator

We have previously shown that BioID-MS and AP-MS can detect high-confidence interactors and identify human transcription factors (Göös et al, 2022) and employed these to study the ETCHbox family member LEUTX (Gawriyski et al, 2023). Here, we applied our MAC-tag-based BioID-MS method (Liu et al, 2018; Liu et al, 2020) for ARGFX, DPRX, TPRX1, TPRX2 and CRX, OTX1, and OTX2 in Flp-In™ T-REx™ 293 cells. All experiments were performed in four replicates (two biological and two technical replicates), and the stringent filtering strategy, coupled with the BioID GFP contaminant database, was used to identify high-confidence interacting proteins (HCIPs). 513 BioID high-confidence interactions (HCIs, BFDR < 0.01) (Dataset EV3), and 72 AP-MS HCIs for the 7 TF bait proteins were identified (Dataset EV4). Of the 513 detected BioID HCIs, 246 were with unique HCIs, and 69 HCIs in the AP-MS. Hierarchical clustering of the BioID HCIs showed that OTX1, OTX2, CRX, TPRX2, and LEUTX cluster together, and are most distant from DPRX (Fig. 2A). Using the BioID-MS HCIs we employed MS microscopy (Liu et al, 2018; Liu et al, 2020), which utilizes the quantitative interactome profiles of localization markers as reference to determine cellular distribution of the bait protein. All tested baits, including LEUTX (Gawriyski et al, 2023), showed a primary localization to 'chromatin' (Fig. 2B), consistent with them being TFs. We compared our list of HCIPs to expression data from

preimplantation embryos and found expression (RPKM > 1) of 227 (Yan et al, 2013) of the 246 unique BioID HCIs between the oocyte and morula stages (Fig. 2C).

Gene Ontology analyses of the combined HCIs from all baits revealed that the interactors of the studies proteins were enriched for processes (GO:BP, FDR < 0.05) such as 'regulation of transcription from RNA polymerase II promoter (GO:0006357)' (101 unique contributing preys), 'chromatin remodeling (GO:0006338)' (17 preys), 'histone modification (GO:001570)' (11 preys), and several cell cycle-related terms (Fig. 2D, Dataset EV5).

We examined the commonly enriched protein complexes that ETCHbox proteins were associated with using the protein complex database CORUM (Tsitsiridis et al, 2022) and found high enrichment of 'Multisubunit ACTR coactivator complex', 'P300-CBP-p270 complex' and 'RSmad complex' (Fig. EV2A and Dataset EV6), and enrichment of several other chromatin modifying complexes more specific for individual baits. For example, the CRX (adjusted p-value = 1.89E−08) and the TPRX2 (adjusted p-value = 4.53E−12) interactomes show strong enrichment of the 'E2F6 complex' (Fig. EV2A).

Of the cellular signaling pathways (reactome.org) associated with most ETCHbox proteins, we detected enrichment of "Activation of HOX genes during differentiation" (Fig. EV2B) except for DPRX. HOX binding motifs are enriched upstream of genes involved in embryonic development (Mallo et al, 2010). The enrichment of "Activation of HOX genes during differentiation" arises from interactions with a group of transcription factors (PAGR1, NCOR1, PAXIP1, NCOA6, RBBP5, NCOA3) and chromatin modifiers (KMT2D, KMT2C, EP300, CREBBP, KDM6A). The most significant enrichment, by FDR, was for "chromatin organization and chromatin modifying enzymes", detected for CRX (26 proteins, FDR = 4.82E−22), OTX1 (9 proteins, FDR = 1.98E−07), OTX2 (16 proteins, FDR = 2.55E−14), and TPRX2 (32 proteins, FDR = 1.23E−26) (Fig. EV2B).

Of the individual interactions identified (BioID), we found CRX to interact with nuclear receptors NR2C1 and NR2C2, nuclear receptor coactivators NCOA1, NCOA2, and NCOA3, and the nuclear receptor corepressor, NCOR2. CRX has previously been reported to interact with a photoreceptor-specific nuclear receptor (Peng et al, 2005). Interactions with nuclear coactivators and repressors are a common theme among the ETCHbox proteins, particularly with OTX1, OTX2, CRX, TPRX1, and TPRX2. The LEUTX interactome (Gawriyski et al, 2023) also identified interactions with the coactivators and repressors listed above, and with NR2C1 and NR2C2. The most enriched Gene Ontology Molecular Function term was "nuclear receptor binding" for ETCHbox and OTX interactomes, except for DPRX (Fig. EV2C).

Consistent with the in silico TAD-domain prediction, we identified BioID-MS HCIs with the KIX-domain containing

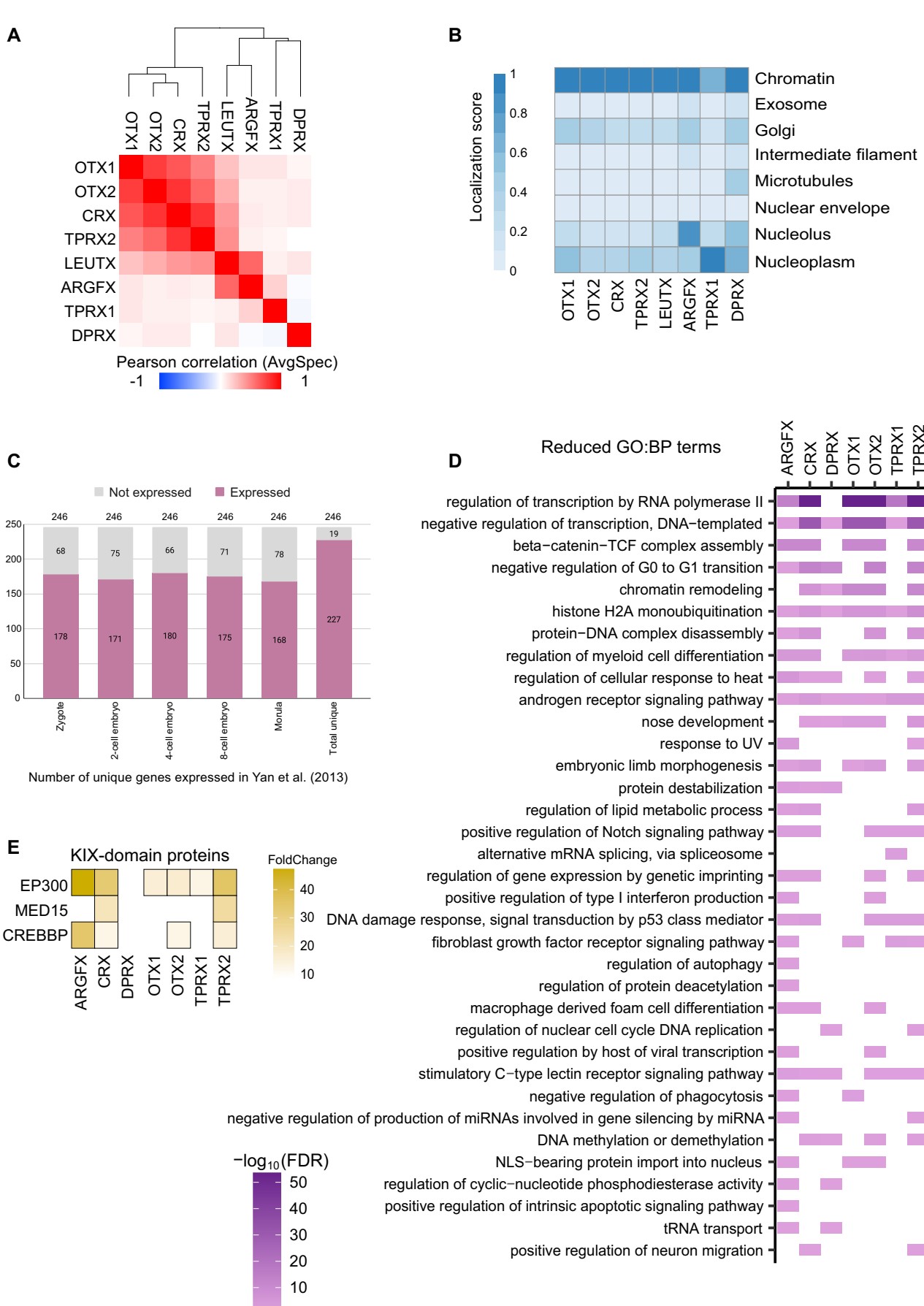

 **Figure 2. Correlation analysis and localization of ETCHbox and Otx family proteins.**

(A) Correlation analysis and clustering of the ETCHbox and Otx family proteins based on their statistically significant PPIs. The LEUTX interactome (Gawriyski et al, 2023) is included in the clustering analysis. Red indicates higher Pearson correlation and blue indicates lower correlation. The phylogenetic tree is built using Ward's algorithm with Euclidian Distance and the ProHits-viz tool. (B) Localization of ETCHbox proteins based on MS-microscopy of BioID PPIs, with a focus on localization to chromatin. Heatmap color indicates localization score between 0 and 1. (C) Expression of BioID-MS prey proteins in embryonic transcriptomics datasets. 227 preys are expressed at RPKM > 1 in Yan et al (2013) in at least one timepoint between zygote and morula cell stages. (D) Gene Ontology biological process (GO:BP) enrichment analysis of the ETCHbox and OTX family baits. Gene Ontology terms are reduced to the highest order term using redundancy based on semantic similarity, and the Log2 FDR indicated in color from high to low magenta-yellow-cyan (FDR < 0.05). (E) Interactions of ETCHbox proteins with known KIX-domain-containing proteins, indicated by the fold change in spectral count in bait samples compared to controls.

proteins MED15, EP300, and CREBBP (Fig. 2E). MED15, the Mediator complex component, interacts with CRX and TPRX2; EP300 interacts with ARGFX, CRX, OTX1, OTX2, TPRX1, and TPRX2; and CREB-binding protein interacts with ARGFX, CRX, CPHX1, OTX2, and TPRX2 (Fig. 2E). None of the KIX-domain containing interactors were found for DPRX, which lacks a putative TAD (Fig. 2E). TPRX2 is currently annotated as a pseudogene, but our results suggest that TPRX2 produces an active and functional protein, consistent with our previous demonstration that TPRX2 is active in preimplantation (Töhönen et al, 2015). Thus, we suggest that TPRX2 is a transcriptional activator and preimplantation factor. This is in line with previous research showing TPRX1 and TPRX2 to upregulate EGA genes (Zou et al, 2022).

## Potential functions for the preimplantation TF interaction network

Our results suggest that the ETCHbox proteins likely function as transcription regulators during preimplantation development. To further understand TFs active in this stage, we expanded our bait set to include 12 TFs that are also active during preimplantation development. These included the PRDL factors CPHX1, CPHX2, DUXA, DUXB, GSC, PITX1, and PITX2 (Töhönen et al, 2015; Liu et al, 2019); EGR2, ZNF263, and KLF3 (Leng et al, 2019); DPPA3 (Gawriyski et al, 2023); and ZSCAN4 (Töhönen et al, 2015; Hendrickson et al, 2017). To further expand the functional interaction networks and to validate the identified ETCHbox interactions, we performed reciprocal analyses for SATB1 and SATB2, both of which were identified as common interactors of the analyzed ETCHbox's and are expressed during preimplantation (Yan et al, 2013).

We performed the same interactome analyses for this second set of these 12 TFs and combined the results with the earlier nine TFs. For this combined set of 21 preimplantation factors, we detected in total 1168 HCIs with BioID-MS. 304 of these interactors were only detected with BioID-MS, 41 were detected with both BioID- and AP-MS, and 67 were only seen with AP-MS (Fig. 3A,B). In most cases, BioID-MS was superior in identifying HCIs compared to AP-MS (Datasets EV3 and EV4), in agreement with our earlier large-scale TF interactome analysis (Göös et al, 2022).

We were also able to capture stable HCIs (with AP-MS) for the 21 preimplantation factors studied. For example, CPHX1 has a stable interaction with the transcriptional repressor and apoptosis promoter TRIM27 (Dataset EV4). DPPA3, a pluripotency (Nakamura et al, 2012) and 8CLC marker (Mazid et al, 2022; Taubenschmid-Stowers et al, 2022; Yoshihara et al, 2022), and a target gene of LEUTX (Gawriyski et al, 2023), stably interacts with the E3 ubiquitin protein ligases UHRF1 and UHRF2 (Dataset EV4).

These findings are consistent with previous studies showing that DPPA3 displaces UHRF1 from chromatin binding, thereby inhibiting UHRF1-mediated methylation and leading to global passive demethylation (Mulholland et al, 2020). TPRX2 stably interacts with multiple members of the SWI/SNF complex, and CREBBP and ZSCAN4 stably interact with the embryonic TF ZNF217.

To focus on TF interactions, we carried out further interaction analyses using BioID-MS. Hierarchical clustering of the HCIs of the 21 potential EGA factors showed that the majority of the K50-class homeodomain factors cluster together (Fig. 3A; Cluster #1). The clustering shows low variance between GSC, PITX1, PITX2, CPHX1, CPHX2, OTX1, OTX2, EGR2, CRX, TPRX2, and KLF3, likely due to these 11 K50-class homeodomain factors having rich interactomes and interacting with many components of chromatin modifying complexes and transcriptional coactivators (Fig. 3A and Appendix Fig. S2A).

The majority of the HCIs observed for the 21 potential EGA factors are previously unreported in protein-protein interaction databases (BioGrid, IntAct, BioPlex, CellMap, String, and PINA) (Fig. 3C). Seven of the 21 factors have no previously detected interactions, and for the other fourteen we report 773 previously unreported high-confidence protein-protein interactions. Majority of the previous studies were antibody-based or other non-MS studies, therefore our analysis describe the first MS analyses for these baits and provides more comprehensive interactomes than those identified by antibody-based methods.

We studied the enrichment of protein complexes in the HCIs of each bait protein using the CORUM database (Giurgiu et al, 2019). "Multisubunit ACTR coactivator complex" is significantly enriched in 12 bait interactomes (FDR < 0.05) (Figs. 3D and 4A; Dataset EV6). Several components of the "SWI/SNF complex", along with other chromatin-modifying complexes (such as the CtBP and RSmad complexes) are enriched for several of baits (Fig. 4A; Dataset EV5). Although we cannot assess whether these interactions are with the active form of these complexes, or obtain interaction stoichiometries, this information can be used to establish the molecular context in which the bait proteins operate. For example, we detect a large number of chromatin-modifying complexes, consistent with bait protein transcriptional regulatory activity. The enrichment of several chromatin-modifying complexes suggests that many of the proteins are DNA bound in transcriptionally active chromatin regions, and the bait proteins do primarily localize to chromatin (Appendix Fig. S2B,C). Several types of zinc finger domains are enriched in the interactome of the baits (Fig. 4B), as are many DNA-binding domains (e.g. ARID, Homeobox). Also, our data indicate the TPRX2 and CPHX2 pseudogenes encode for functional DNA-binding proteins that do seem to bind transcriptionally active regions.

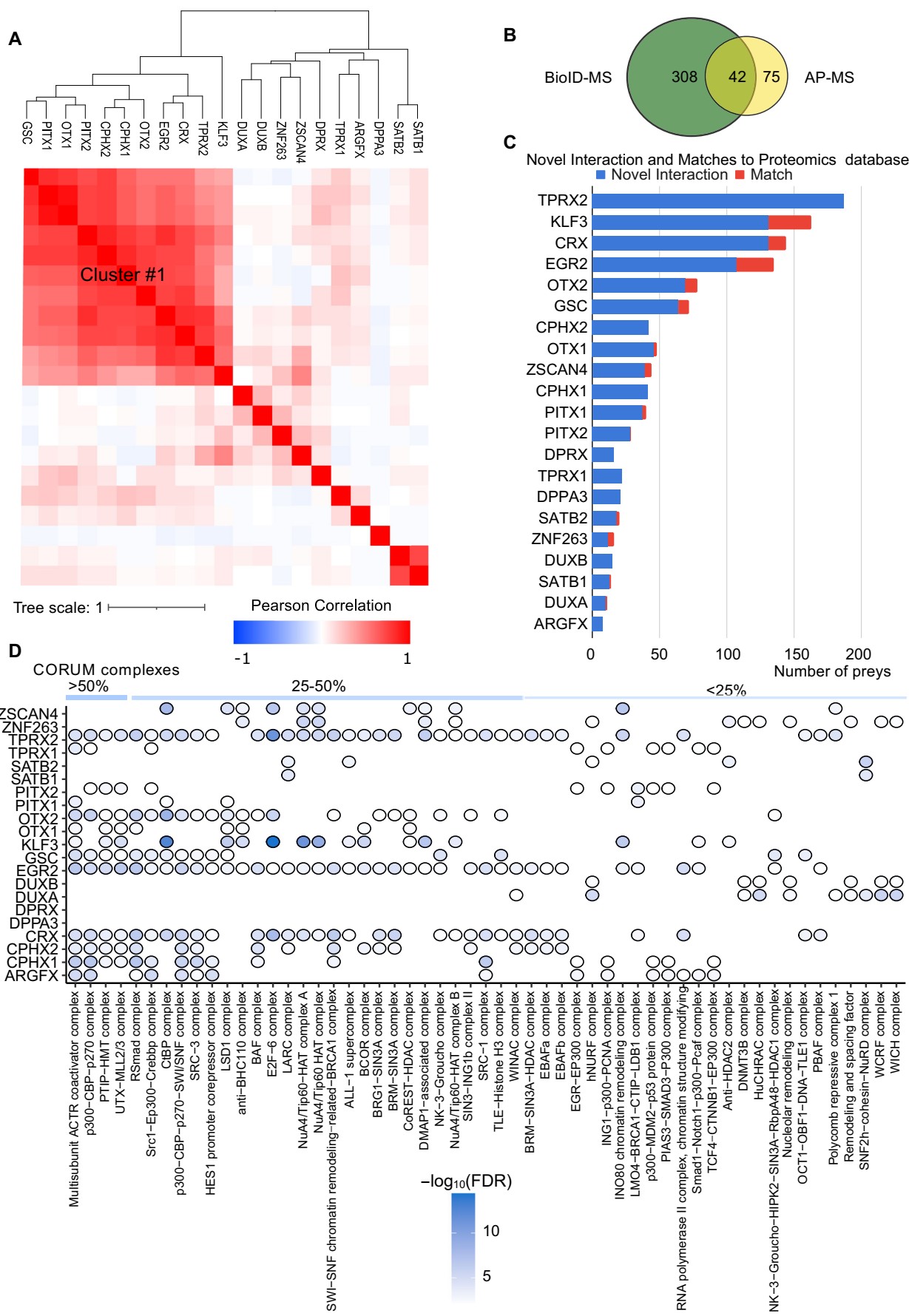

**Figure 3. Correlation analysis and enrichment of protein–protein interactions.**

(A) Correlation analysis and clustering of all 21 baits analysed in this study based on their high-confidence PPIs. The phylogenetic tree is derived using Ward's algorithm with Euclidian distance, with a scale of 1 depicted on top of the correlation clustering. AverageSpec (average PSM) is used as an abundance term for the correlation analysis, and the tree is made using the ProhitsViz tool. Cluster #1 is a group of studied baits that share the largest number of interactors. (B) Total number of unique BioID-MS (green) and APMS (yellow) interactions found in the dataset and the overlap between the datasets. (C) Number of novel PPIs identified in our analyses compared to known PPIs in proteomics databases. Seven proteins had no previous information in interactomics databases (e.g. presumed pseudogenes TPRX2, CPHX2). (D) Significantly enriched (FDR < 0.01) protein complexes in the interactomes of each bait. Protein complexes were obtained from the CORUM database. Only complexes enriched in more than two baits are shown. Increased blue color indicates inverse statistical significance ($-\log_{10}$ (FDR)). Complexes are ordered by frequency.

Of the 347 HCIPs, 286 (82%) are expressed in >90% of human tissues in ExpressionAtlas, suggesting that they are general cellular factors (Appendix Fig. S2D) (Papatheodorou et al, 2018). 18 HCIPs were not detected in any of the somatic tissue types in ExpressionAtlas, suggesting their expression is induced by the expression of the studied TFs. The majority of these HCIPs are TFs or transcriptional cofactors: TEAD1, TLE5, ZNF462, TCF20, MITF, SOX13, TBX2, ZNF423, and DACH1; and epigenetic modifiers: MECOM, PRDM16, SMARCA1, RESF1, ZZZ3, BAP18, and MIDEAS. TEAD1, ZNF462, TCF20, MITF, SOX13, MECOM, PRDM16, ZZZ3, and MIDEAS are expressed in preimplantation embryos between the oocyte and morula stages (RPKM > 1 in at least one cell stage; Yan et al, 2013). That these factors are not detected in somatic tissues and are identified as HCIPs may be indicative developmental function and gene regulatory relationship between the bait and the HCIP.

Most of the studied EGA factors function in chromatin modification and/or epigenetic regulation, given their interactomes. This is reflected in the GO:MF enrichment (FDR < 0.05) of "acetyltransferase activity" for all but four baits (Fig. EV3A), GO:BP enrichment (FDR < 0.05) of terms such as "regulation of histone acetylation", "chromatin-mediated maintenance of transcription" (Fig. EV3B), and Reactome pathway enrichments where commonly enriched terms (FDR < 0.05) include "Chromatin organization" and "Chromatin modifying enzymes" (Fig. EV4). Several developmental terms are also enriched in GO:BP (e.g. "positive regulation of developmental process"), and Reactome (e.g. "Activation of HOX genes during differentiation"), consistent with the proposed function of these factors (Figs. EV2B and EV4). Comparing the identified HCIs to other developmental pluripotency factors and revealed that many of the interactomes are enriched for the same factors as pluripotency factor interactomes (TCFCP2L1, POU5F1, CTNNB1, and ESRRB) (Appendix Fig. S3A).

Of the 347 unique BioID-MS HCIPs 325 (92.5%) are found expressed in HEKs in ExpressionAtlas (Appendix Fig. S3B) (Papatheodorou et al, 2018). Of the 347 unique BioID HCIPs, genes for 249, 262, 246 are expressed, respectively, in a published embryonic RNA-Seq datasets (Yan et al, 2013; Petropoulos et al, 2016; Zou et al, 2022), in the 8-cell stage (limit of detection RPKM > = 1, RPKM > = 1, FPKM > 1, respectively), (Appendix Fig. S3C), whereas in the ICM stage 244, 247, and 250 are expressed in the ICM stage, respectively (Appendix Fig. S3D). Majority of BioID-MS HCIPs are detected expressed in the embryonic transcriptomic datasets as well as HEK293s (Appendix Fig. S3C,D). The proportion of BioID-MS HCIPs expressed in embryonic datasets remains relatively proportionate regardless of dataset (Appendix Fig. S4A). Comparison with 8CLCs datasets reveals 39 genes considered hub genes by Mazid et al, 2022 and 19 genes considered key 8CL-genes by Yoshihara et al, 2022 (Appendix Fig. S4A). Many of the genes considered as key 8CLC-

genes are also expressed in HEK293 (Appendix Fig. S4B). Three of the highlighted 8CL-genes that are also detected as BioID-MS HCIs are not detected in HEK293 in the ExpressionAtlas (Appendix Fig. S4C). These BioID-MS HCIPs are TRERF1 (interacts with TPRX2, EGR2, CRX), ZNF423 (interacts with KLF3), and HRNR (interacts with DPRX, CPHX2) (Appendix Fig. S4C).

## Preimplantation interaction network hub proteins

The 1186 BioID HCIs map to 364 unique proteins, and therefore the majority of the detected HCIs were detected in the total interactome of several baits. The most common shared HCIs are MIDEAS and ZFHX4, which interact with 14 baits. EP300, PRR12, TRPS1, and ZFHX3 interact with 13 baits. DNTTIP1, KDM2B, KMT2C, KMT2D, NCOR2, and PAXIP1 and several ZNF-factors interacted with 11 baits (Fig. 5A). Many of these shared HCIs are TFs, cofactors or chromatin modifiers especially lysine methyltransferases, and Notch signaling (Fig. 5A), and the majority of identified HCIs are shared in the network (Fig. 5B). Gene Ontology enrichment of the preys shared by >1 bait indicated their role in transcriptional regulation via RNA polymerase II (Fig. 5C).

Given the link to transcription regulation, we compared our BioID data to the BioID data of a representative set of >100 human TFs (Göös et al, 2022), to determine if any preys were specifically enriched in our EGA data compared to the more general set of TFs. We performed Fisher's exact test to compare our observed values with the >100 human TF set (Göös et al, 2022), and found a significant difference in the HCIs (FDR < 0.05) (Dataset EV7), including ZFHX3, ZFHX4, RERE, SALL1, and PCGF5. The zinc finger homeobox proteins ZFHX3 and ZFHX4 are TFs expressed in preimplantation embryos (Yan et al, 2013); RERE is a developmental transcriptional repressor and apoptosis factor; and SALL1 and PCGF5 are transcriptional repressors. Conversely, our data was significantly under-enriched for corepressor BCOR and SWI/SNF complex member PBRM1 (Dataset EV7).

To further compare our dataset with previously published embryonic transcription factor BioID-MS datasets we combined data from our work on LEUTX (Gawriyski et al, 2023) and DUX4 (Vuoristo et al, 2022) and performed clustering analysis (Appendix Fig. S5A). Clustering analysis shows DUX4 and LEUTX interactomes exhibit a higher degree of similarity to each other and more akin the major cluster in Fig. 3A (Appendix Fig. S5A). In Gawriyski et al, 2023, LEUTX was found to interact directly with EP300 and CREBBP which is reflected as very strong interaction also through BioID-MS (Appendix Fig. S5B). CORUM enrichment analysis of the expanded bait set shows LEUTX and DUX4, both known as potent transcriptional activators, to share many chromatin-modifying complex interactions with majority of the baits (Fig. EV5A). The InterProDomain protein domain of prey

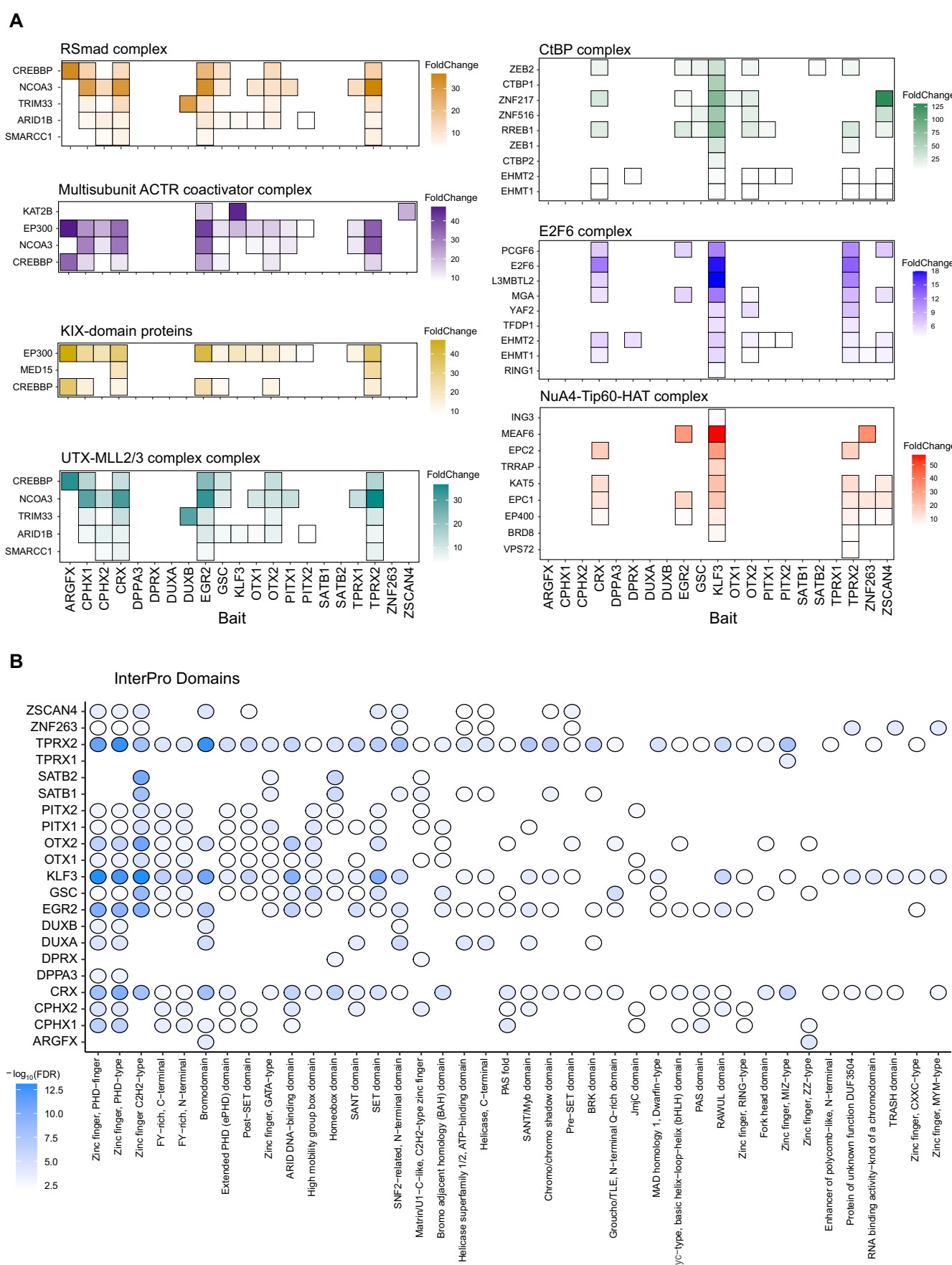

**Figure 4.  Analysis of high-confidence interactions and enriched domains.**

(A) The number of high-confidence interactions and their abundance for the key complexes RSmad, CTBP, E2F6, ACTR, UTX-MLL2/3, and NuA4-Top60-HAT, and predicted key targets (KIX-domain-containing proteins) for all 21 baits. Color indicates fold change between samples and controls. (B) Statistically significant enriched InterPro domains (FDR < 0.05) in the interactome of each bait, indicated by the number of shared baits. Color indicates −log$_{10}$ (FDR) of enrichment analysis. Domains are ordered by −log$_{10}$ (FDR) and frequency.

enrichment analysis shows DUX4 to be more akin to the majority of baits produced for this paper (Fig. EV5B)

## Validation of BioID-MS HCIs in iPSC model and co-expression co-immunoprecipitation

Despite the insights provided by the preimplantation interaction network, our interactome data could contain false-positive interactions without biological relevance. To assess this and validate the detected interactions we performed the BioID-MS for several bait proteins in human induced pluripotent stem cell line (iPSC) HEL24.3. We successfully cultured and modified the cells by introducing the MAC3-tagged (Salokas et al, 2022) transcription factors via electroporation and subsequent selection protocols. We obtained cell lines constitutively expressing MAC3-tagged DUXA, TPRX2, and ZSCAN4. The interactomes obtained from iPSC cell lines showed high overlap with the BioID-MS HCIs produced in the Flp-In™ T-REx™ 293 cell line (Appendix Fig. S6A). Notably, the DUXA interactome overlap was at a prominent level of 91.7% (Appendix Fig. S6). Similarly, TPRX2 and ZSCAN4 overlaps were 69.1% and 84.4%, respectively (Appendix Fig. S6A). The difference in the overlap with TPRX2 most likely results from the better sensitivity of the Flp-In™ T-REx™ 293 model, as all the Flp-In™ T-REx™ 293-specific interactions are linked to regulation of transcription (Appendix Fig. S6B).

We performed additional PPI validation to assess the accuracy of our interaction dataset. For this, 96 protein interaction pairs (detected as HCIP with >20 PSM values, suggesting a direct interactor) were selected and validation experiment performed using co-expression co-immunoprecipitation (co-IP) dot blotting (Appendix Fig. S7; Dataset EV8). The validation ratio was 82% (79/96 of the tested interaction pairs validated). In sum, the results from both validation methods suggest that the interactions of the preimplantation interaction network are reproducible and also biologically relevant.

## TF binding to developmental promoter regions

Our BioID-MS data suggest the ETCHbox and other EGA-implicated proteins to have transcription-related roles, but their specific functions remain unclear. To gain insight into these specific functions, we analyzed these proteins' genomic binding patterns, using chromatin immunoprecipitation sequencing (ChIP-Seq). We analyzed ARGFX, CPHX1, CPHX2, CRX, DPRX, DUX4, DUXA, DUXB, EGR, KLF3, TPRX1, TPRX2, and ZNF263 in Flp-In™ T-REx™ 293 cells, using the MAC-tag approach (Liu et al, 2020; Liu et al, 2023). MAC-tagged baits were induced for 24 h and collected. We identified 187,090 statistically significant ChIP-Seq peaks in total (MACS2, FDR < 0.05). CPHX1 and KLF3 mostly bind to proximal promoters (defined as ChIP-Seq peaks that map to 0–1 kb from transcription start site (TSS) from GENCODE annotation), whereas CPHX2, DUX4, DUXA, DUXB, and TPRX2 map to

intergenic regions and introns and show very little proximal promoter binding (Fig. 6A). We checked whether peaks identified overlapped with known enhancers, as identified by the FANTOM5 project (Lizio et al, 2015) and found that EGR2 binds the most known enhancers (9.8% of peaks) (Fig. 6B).

We identified the predicted primary motifs of the binding regions in all datasets using the MEMESuite MEME-ChIP de novo motif analysis tool. In the case of CPHX1 (14,299 peaks) and CPHX2 (866 peaks), motif finding top hits were CTCF and TEAD3 motifs, respectively. For KLF3, ZNF263, EGR2, DUX4, and DUXA, we identified a motif matching their previously reported motifs in the JASPAR database (Fig. 6C) (Castro-Mondragon et al, 2022). For DUXB, we could not find a previously known human binding motif in JASPAR and instead performed two de novo motif finding approaches (Fig. 6C). A 36 bp motif, consisting of a partial Alu element, has previously been found to be enriched upstream of genes involved in EGA (EEA-motif) (Töhönen et al, 2015). We found partial hits to the EEA motif as the top binding motif hit in CRX, DPRX, TPRX1, and TPRX2 (Fig. 6C). The EEA motif contains a binding site for K-50 class homeodomains (TAATCC/GGATTA), with identified de novo motifs expanding out to match the EEA-motif mostly by matching the 'GCTGGGATTACA' core region of the EEA-motif (Fig. 6C). Conversely, de novo motif identification identified the EGR2 motif centrally enriched in the EGR2 dataset, with the second most-enriched motif being a partial EEA motif.

We studied the overlap of the identified ChIP-Seq peaks with repetitive elements using HOMER annotatePeaks and found that the majority of the baits' binding sites are enriched for various simple repeat elements and rRNA elements (Fig. 6D). Consistent with DUX4 binding MaLR elements (Vuoristo et al, 2022), we found that DUX4 binding sites are enriched for several ERVL-MaLR elements, some of which are shared by CRX and TPRX2. CRX binding sites are enriched for SINE MIR elements and LTR5_Hs elements, and TPRX2 binding sites are enriched for Alu elements, consistent with another family member, LEUTX, binding Alu elements (Gawriyski et al, 2023).

To search for motifs enriched proximal to the EEA-motif in the PRDL datasets, we used the spaced motif finding tool SpaMo (MEMESuite). The EEA-motif is frequently close to the ATF3, MITF, nuclear receptor (NR1H3, NR1I2, NR2C1), NFIA, E2F6, and KLF3, and EGR2 motifs. Finding proximally enriched nuclear receptor binding motifs and E2F6 complex member binding motifs is consistent with our BioID results, in which we found enrichment of the E2F6 complex members and nuclear receptors (NR2C1, NR2C2) and several nuclear receptor cofactors (NCOA1, NCOA2, NCOA3, NCOA6, NCOR1, NCOR2) as HCIs for several baits. For CRX, we found four spatially enriched motifs that correspond to BioID-MS HCIs E2F6, NR2C1, NFIA, and TFAP4 that are also BioID-MS HCIs.

To validate our ChIP-Seq data, we compared our ChIP-seq peaks for EGR2 and ZNF263 to corresponding ChIP-Seq data from

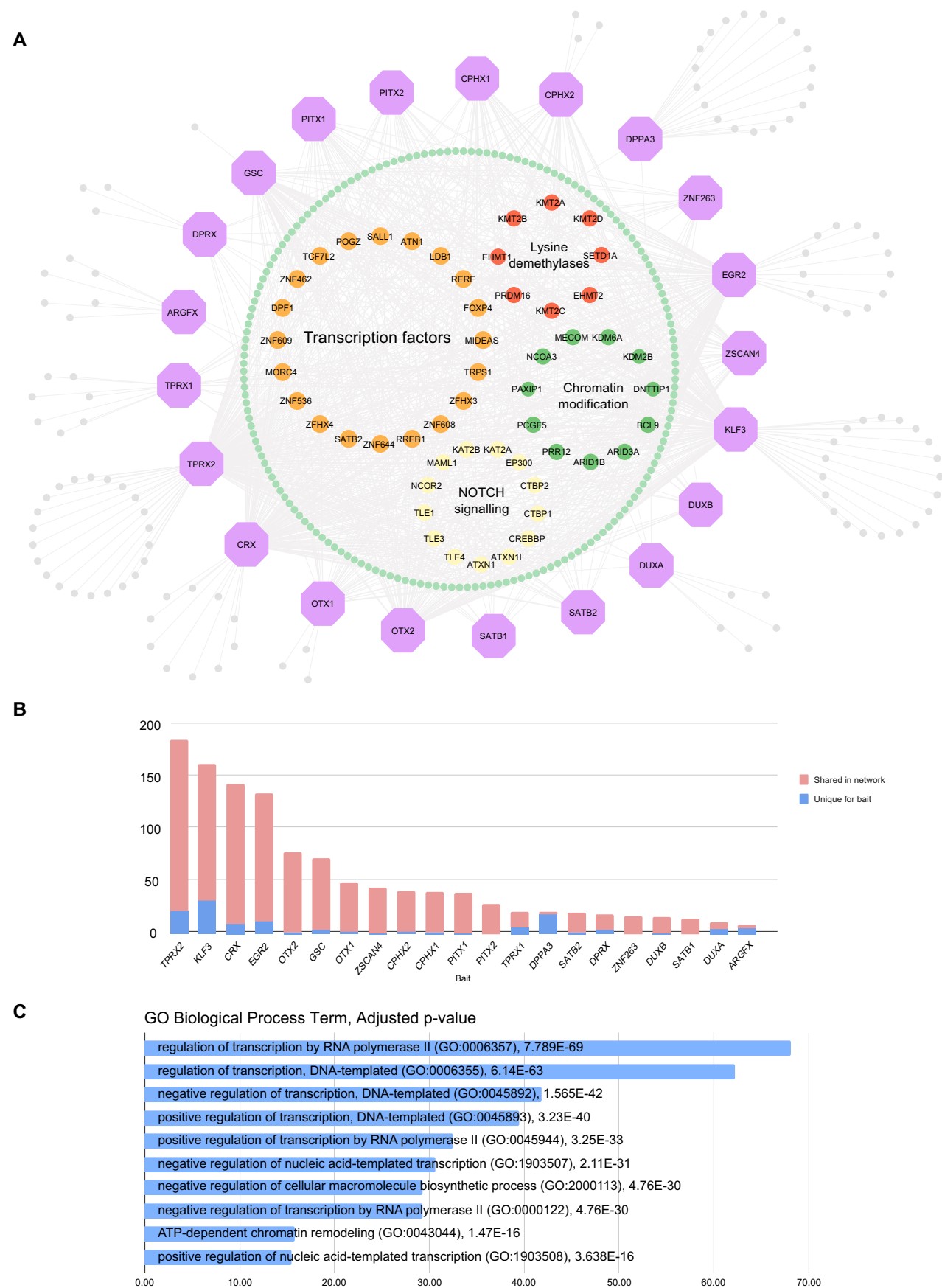

**Figure 5. Network analysis of high-confidence interactions.**

(A) The network of all interactions is presented such that unique high-confidence interactions (HCIs) per bait are clustered on the outside ring and shared HCIs are placed in the inner ring. The preys with the most shared baits are highlighted in the middle based on their known function. (B) Number of HCIs that are shared in the total network compared to the number of HCIs unique to the bait in the total dataset. (C) Top 10 Gene Ontology biological process enrichment analysis results of all shared network HCIs (HCIs that can be found as HCIs for at least two baits) (Fisher Exact Test, FDR < 0.05).

the ENCODE project. We compared our KLF3 data to KLF4 ENCODE data, as KLF3 data is not present in ENCODE. We observed good overlap, with 63% of peaks we identified for EGR2 being present in the ENCODE data, and 71% of peaks in the ENCODE data present in our ZNF263 dataset (Fig. 6E), verifying our data are consistent with previous findings. For KLF3 to KLF4 comparison we found 43% of our peaks overlap KLF4 ENCODE peaks (Fig. 6E).

The ChIP-Seq peaks showed some overlap between proteins (Fig. EV6A). For example, 51% of TPRX1 peaks overlap with CRX peaks, and 54% of TPRX2 peaks overlap with CRX peaks (Appendix Fig. S7A). Overlap of ChIP-Seq peaks is observed in some key regions that peak in expression in the 8-cell stage (Fig. EV6B and Appendix Fig. S3B). For example, high levels of binding are observed in the D4Z4-repetitive region, which contains the DUX4 gene (Fig. 7A). *DUX4* is upregulated at the 2-cell stage and rapidly downregulated and silenced at the four-cell stage and thereafter (Hendrickson et al, 2017). We found binding in this region by CPHX1, DPRX, CRX, EGR2, KLF3, DUXB, and ZNF263 which indicates them participating in regulating the *DUX4* genomic region. Similarly, high levels of binding of CPHX1, CRX, DPRX, EGR2, KLF3, TPRX2, ZNF263 in the TPRX1-CRX-TPRX2 genomic locus that is open the 8-cell stage indicates these genes participating in the regulation of also this region (Fig. 7A). Binding of CPHX, DPRX, EGR2, KLF3, and ZNF263 to the *ZSCAN4* promoter region that peaks in expression in the 8-cell stage and the binding of DUX4 to the second exon indicates that these proteins participate in the transcriptional regulation of *ZSCAN4*. We found similar results for *DUXA* promoter region where DPRX, EGR2, KLF3, and CRX are bound, suggesting these proteins regulating the expression of *DUXA* (Fig. 7A). We found ZZZ3 as a strong BioID-MS interactor for KLF3. ZZZ3 is expressed at low levels early in embryogenesis and its expression peaks at the 8-cell stage (Yan et al, 2013). There are binding sites for EGR2, KLF3, ZNF263, and CPHX1 at the TSS and proximal promoter region of *ZZZ3* (Fig. 7A). Further, *KLF5* and *CTCF* regulatory regions show binding from the studied EGA TFs in line with findings by Zou et al (2022) (Fig. EV6B).

We investigated the function of genes that reside in proximity to our identified binding sites by performing genomic region GO term analysis, using ChIPSeeker (only peaks bound +/−3 kb from a known GENCODE TSS were considered, FDR < 0.01) (Fig. 7B). Several developmental functions were shown to be enriched in more than one target protein peak list, including "in utero embryonic development", "forebrain development", and "embryonic organ morphogenesis". KEGG Pathway enrichment analysis (FDR < 0.01) on the same proximal gene set showed enrichment of several signaling pathways, such as Wnt, Hippo, and mTOR, and the term "Signaling pathways regulating pluripotency of stem cells" was enriched for CPHX1, DPRX, EGR2, KLF3, TPRX1, and ZNF263. We analyzed which of these proximal genes have been

reported to be expressed (RPKM > 1) in preimplantation embryos (Yan et al, 2013), and identified 9482 unique genes proximal to our binding sites, many of these for more than one bait, for a total of 33,065 observations. Of these 9482 unique genes, 3689 genes peak in expression at the 8-cell stage, consistent with the expression pattern of the TFs we have analyzed.

## Discussion

The functions of the proteins involved in embryonic genome activation (EGA) in the early embryo are little known. Here, we analyzed the interactions and functions of a set of potential EGA transcription factors, with a focus on the ETCHbox gene family. Using in silico domain prediction we found that most of these proteins contain a transactivation domain (9aaTAD), typical of transcriptional activators. This domain facilitates interaction with KIX-domain-containing proteins, such as MED15, EP300, and CBP. We observed an interaction with at least one KIX-domain-containing protein for each of our studied bait proteins with an in silico predicted 9aaTAD, consistent with the presence of a functional 9aaTAD. In the PRDL family we observed high-confidence interacting protein (HCIP) signals for baits CRX and TPRX2 with MED15; EP300 HCIs with ARGFX, CRX, OTX1, OTX2, TPRX1, and TPRX2; and CREB-binding protein HCIs with ARGFX, CRX, OTX2, and TPRX2. No KIX-domain-containing HCIs were found for DPRX. These HCIs, together with the many other chromatin modifying or otherwise transcription-related complexes identified as HCIs, support the idea that these proteins are transcriptional activators that bind actively transcribed genomic regions. That DPRX does not share these interactions is consistent with earlier results suggesting DPRX as a transcriptional repressor (Jouhilahti et al, 2016). TPRX1 and TPRX2 were recently shown by Zou et al (2022) to upregulate several EGA genes and their combined knockdown together with TPRXL, potentially created from reverse transcription of *TPRX1* (Holland et al, 2007), lead to developmental delay and transcriptional misregulation. The *TPRX* genes were shown to have shared regulatory targets and knockdown of *TPRX1* and *TPRX2* alone did not lead to developmental delay (Zou et al, 2022).

CRX has previously been reported to interact with the photoreceptor-specific nuclear receptor NR2E3 (Peng et al, 2005). We found BioID-MS HCIs for CRX with nuclear receptors NR2C1 and NR2C2, as well as the nuclear receptor coactivators NCOA1, NCOA2, NCOA3 and nuclear receptor corepressor NCOR2 (HCIs with NR-cofactors are also found for OTX1, OTX2, TPRX1, and TPRX2). The LEUTX interactome (Gawriyski et al, 2023) also included these coactivators and repressors, as well as NR2C1 and NR2C2. This possible interplay of the ETCHbox family with the nuclear receptors and nuclear receptor cofactors is previously uncharacterized and will require further research.

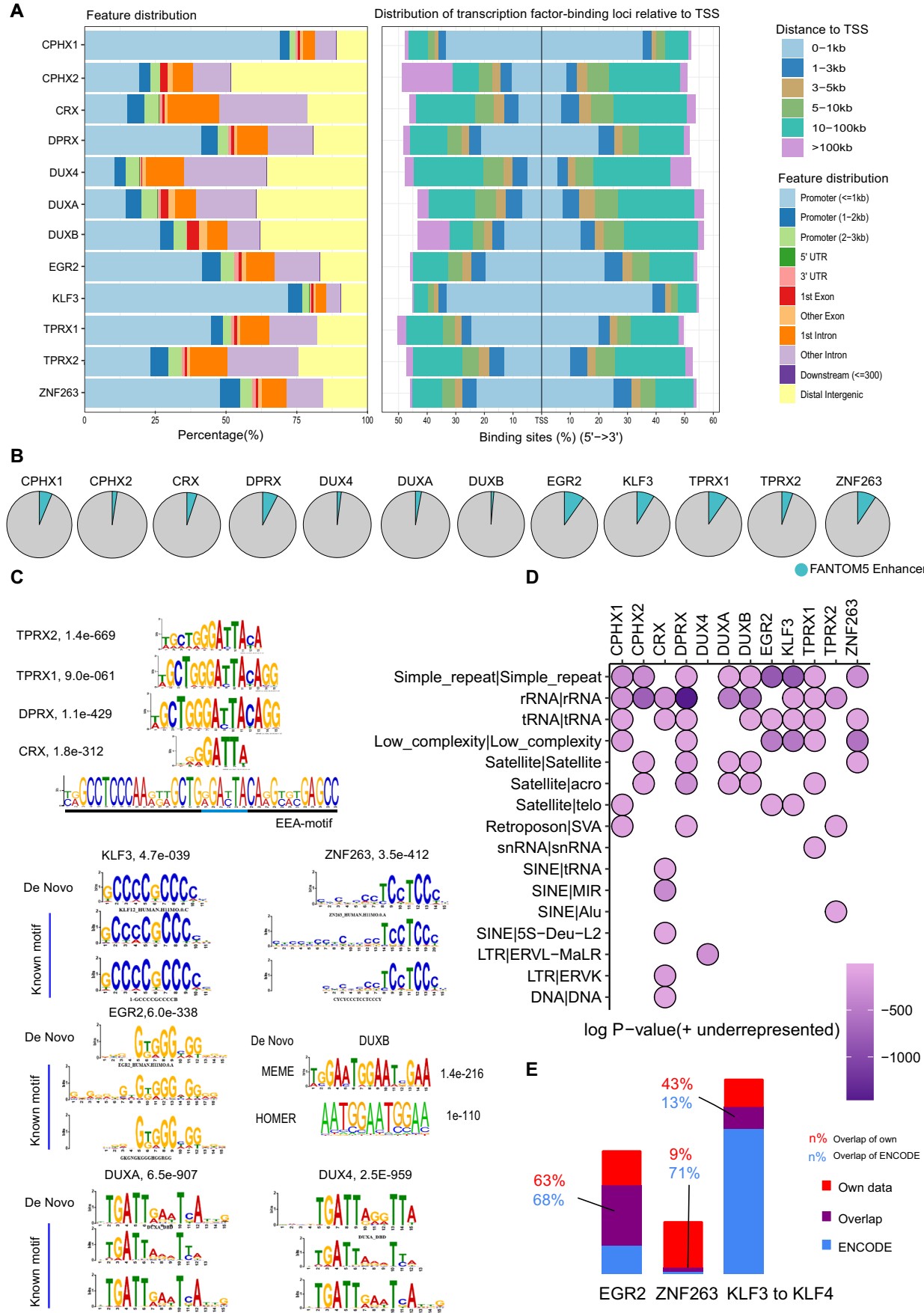

Figure 6.   Analysis of ChIP-seq peaks and motifs.

(A) Annotation of ChIP-seq peaks to known GENCODE transcription start sites (TSS) with a range of +3 kb to −3 kb. Two plots depict the percentage of peaks that annotate to known genomic features and their distance to the TSS for each studied protein. (B) Binding to known FANTOM5 enhancers for each studied protein. Binding is defined as the statistically significant ChIP-seq peak overlapping a known FANTOM5 enhancer (bedtools overlap). The proportion of peaks overlapping FANTOM5 enhancers is depicted in teal color, and the proportion of peaks not overlapping enhancer elements is shown in gray. (C) De novo motif prediction results for significant ChIP-seq peaks using MEMETools MEME-ChIP. PRDL proteins are aligned to the EEA factors identified in Töhönen et al (2015). Other factors are aligned to the known ChIP-seq peak motif profile for that factor, except for DUXB, which had no known motif, so we also show the de novo prediction of another motif-finding tool, HOMER. (D) HOMER annotatePeaks repeat element enrichment analysis results for ChIP-seq peaks of each bait. This plot shows the significantly enriched (HOMER annotatePeaks, FDR < 0.01) major categories of repeat elements. The color is the Log P-value (underrepresented) (filtered < 0) produced by HOMER, a measure that reflects the significance of the enrichment compared to the genomic frequency. Deeper purple indicates higher enrichment in ChIP-seq peaks as compared to genomic frequency. (E) Comparison of the ChIP-seq peaks produced in this study to those published previously in the ENCODE project. We compared the ChIP-seq peaks produced in this study to experiments found in ENCODE for EGR2 and ZNF263 that had previous data published. For KLF3, we compared to KLF4, as KLF3 TF-ChIP-seq was not found in ENCODE. The number of genes unique in the data in this study is shown in red, overlap with ENCODE data is shown in purple, and peaks unique to the ENCODE experiments are shown in blue.

For each bait protein, we found several HCIs with proteins that form chromatin-modifying complexes. Most of the bait proteins interacted with members of the non-canonical polycomb complex E2F6 (PRC1.6) (Attwooll et al, 2005), which is involved in cell cycle control and cell proliferation (Attwooll et al, 2005). HCIs with this complex has also been found for LEUTX (Gawriyski et al, 2023), ZMYM2 (Connaughton et al, 2020), and members of the CoREST complex (Barnes et al, 2022). We also found E2F6 DNA binding motifs enriched proximal to the binding site of many of our bait proteins. TPRX1, CPHX1, DPRX, EGR2, KLF3, and ZNF263 bind close to the E2F6 gene which is rapidly and highly expressed (RPKM = 58) during the 8-cell stage (Yan et al, 2013). Interaction with KIX-domain containing proteins, such as CBP and EP300, and through them, complexes such as RSmad and ACTR coactivator complex, and other chromatin-modifying complexes, such as E2F6 and CtBP, likely explain the transcriptional activity of these TFs. Published BioID-MS analyses on 109 TFs found many TFs interacted with TFIID or SAGA complexes (Göös et al, 2022). We found no significant enrichment of SAGA complex and few TFIID interactions, even though KLF3 proximally interacted with four TFIID subunits. Rather, PRDL factors preferentially interact with KIX-domain-containing proteins, nuclear receptors, and SWI/SNF and E2F6 complexes. A comparison of our BioID-MS dataset with that of Göös et al (2022) revealed significant interaction differences for 21 proteins, including: chromatin modifying proteins or complex members, PCGF5 and INO80E; TFs ZFHX3, ZFHX4, ZNF521, SALL1, and RERE; cofactors BCL9L, ANT1, and DNTTIP1; and proteins of unknown function, such as C15orf39 and MIDEAS. Overall, BioID-MS analysis identified significantly more interactions for each studies bait compared to AP-MS, in line with previous findings (Göös et al, 2022). This is most likely due to the transient nature of the TF interactions. For example, recent studies suggest that the physicochemical properties of the activation domain can control TF assembly at chromatin by driving phase separation into transcriptional condensates (Trojanowski et al, 2022).

We observe that most studied TFs interact with nuclear factors, as reported previously (Göös et al, 2022). While all samples had at least one peptide identification for NFIA, only 6 baits had a statistically significant interaction with NFIA, suggesting that NFIA is a general factor that in many cases is not specific to the function of studied TFs. We found very little interaction with Mediator proteins, similarly as reported previously (Göös et al, 2022). HCIs are observed for CRX and TPRX2 with MED15, but no other Mediator complex members. Stable interaction with Mediator seems to be rare for TFs; we previously observed direct interaction

for the pioneering TF DUX4, using AP-MS (Vuoristo et al, 2022). However, we observed that Mediator proteins often copurify in control samples in BioID-MS experiments. 28 members of the Mediator complex are identified through GFP control baits, for example: MED1 is identified in 92% of all GFP controls and 100% of NLS-tagged GFP controls. Why Mediator often co-purifies in affinity purification experiments is unknown, most likely it reflects the abundancy of the complex.

We found many common HCIs for our baits. For example, the transcriptional corepressor MITF was detected both as a HCI and in spatial motif analysis, and is highly expressed during the oocyte and 4-cell stages, after which its expression is silenced. EGR2, KLF3, CPHX1, and ZNF263 proteins all bind proximally to the MITF gene (+/−3 kb from TSS). MIDEAS is also expressed between the oocyte and morula stages and its TSS is bound by ZNF263, EGR2, CPHX1, and KLF3. We suggest these proteins may be potential targets for stem cell reprogramming methods. As hub proteins we found several lysine methyl transferases (EHMT1, EHMT2, KMT2A, KMT2B, KMT2D, SET1A, PRDM16) which suggest histone methylation as a key function for the here studied bait set.

Using KLF3 as a bait for BioID-MS, we identified ZZZ3 as a strong BioID-MS HCI. ZZZ3 was only identified in the KLF3 BioID-MS and is not commonly expressed in tissues or cell lines. This suggests that KLF3 first upregulates the ZZZ3 gene and then interacts with the ZZZ3 protein. KLF3 binding sites were found in the ZZZ3 TSS region, and ZZZ3 peaks in expression during the 8-cell stage in human embryos during which KLF3 is also expressed (Yan et al, 2013).

We showed that the previously annotated pseudogenes TPRX2 and CPHX2 produce functional protein products. These genes are expressed during the 8-cell stage in human embryos (Töhönen et al, 2015). Analysis of 8CLCs has found TPRX1, ZSCAN4, DUXA, DUXB, ARGFX, PITX2, and GSC as potential markers of 8CLC gene expression (Mazid et al, 2022; Taubenschmid-Stowers et al, 2022). TPRX2 functions as a DNA-binding protein with an interactome that is indicative of a potent transcriptional activator, with prior research possibly misattributing some of TPRX2 functions to TPRX1 due to high sequence similarity and the pseudogene annotation.

Overall, our study reveals key insights into the function of this underexplored protein family, emphasizing their PPIs and binding sites as potential research targets. We validated these interactions through co-IP and confirmed the interactome analysis for three pivotal PRDL transcription factors in induced pluripotent stem cell (iPSC)

## A

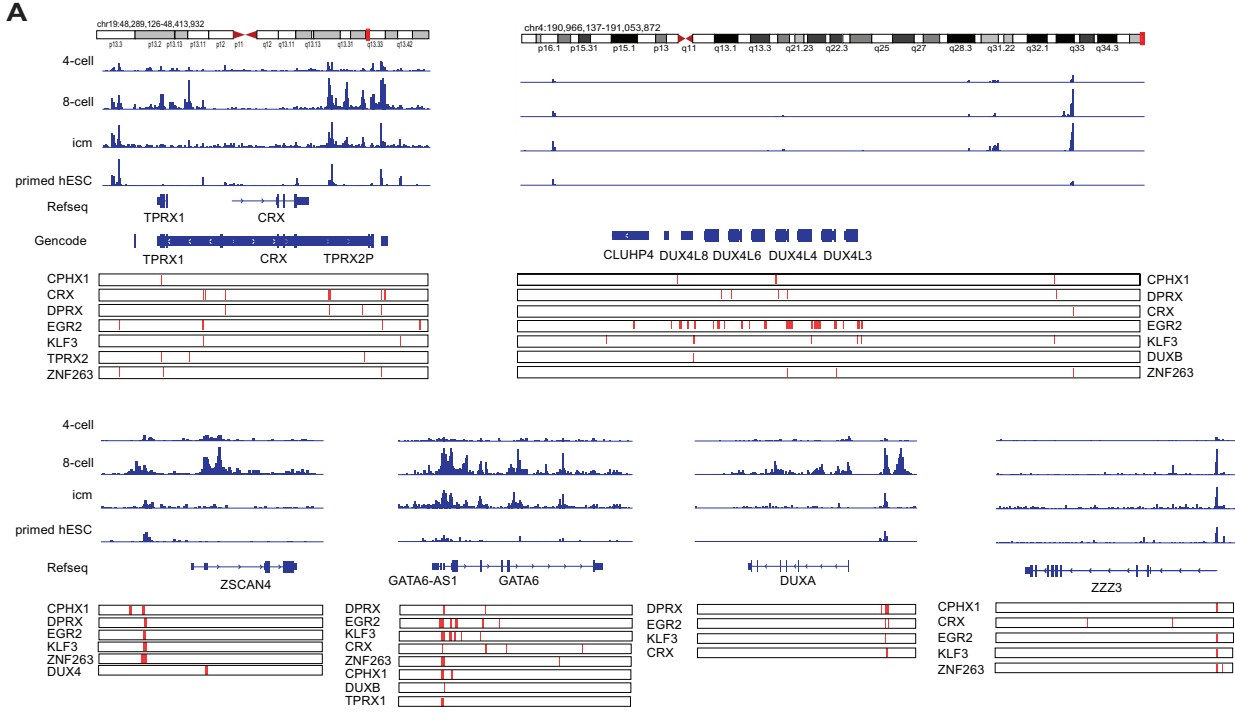

## B

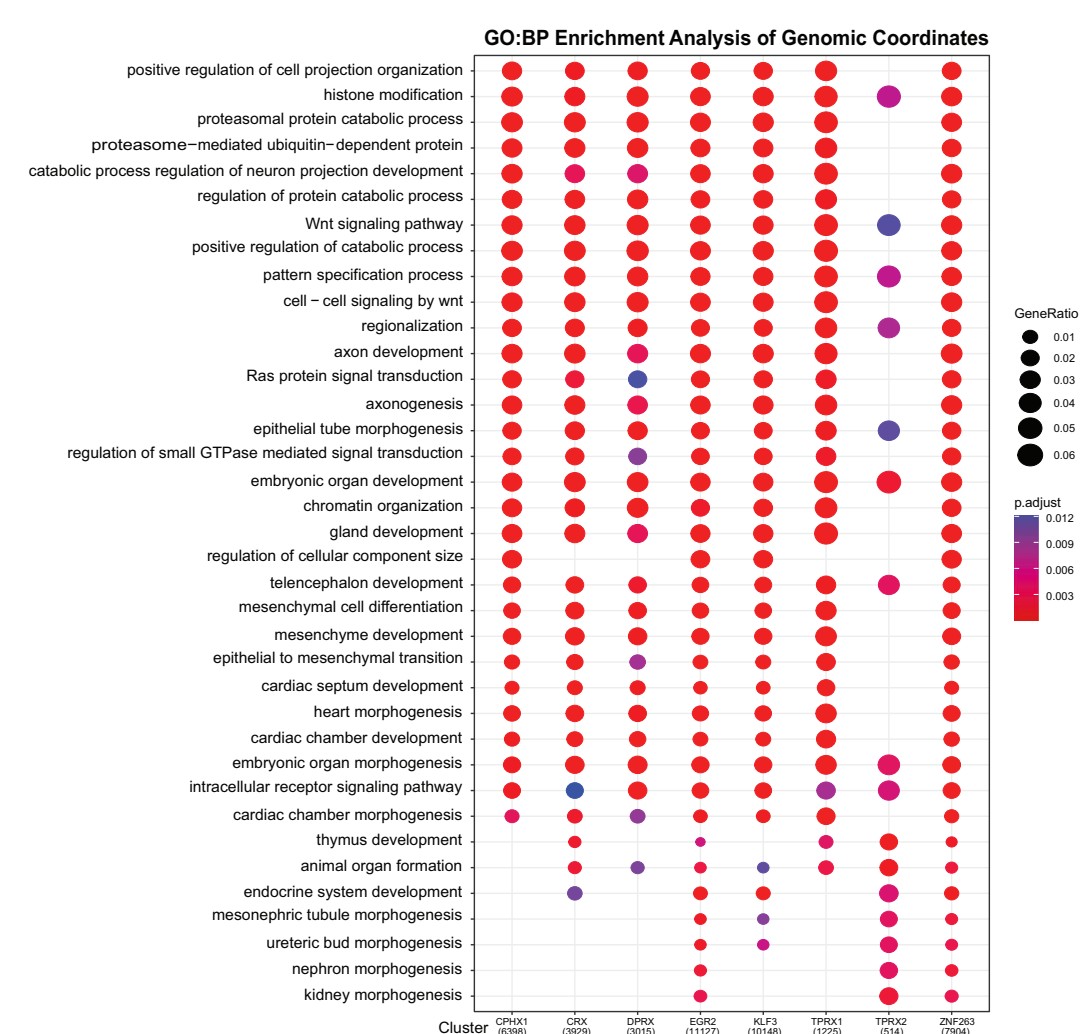

◄ **Figure 7.  Gene ontology and ATAC-seq analysis of ChIP-seq peaks.**

(A) The embryonic ATAC-seq data from Wu et al (2018) from 4-cell, 8-cell, and ICM embryos and primed hESCs over key gene regions identified through our datasets. Genomic locations of differential ChIP-seq peaks are shown in red, relative to their position to key genes (RefSeq annotation). (B) Gene ontology functional enrichment analysis for peaks that bind proximally to genes (−3 kb to 3 kb from TSS). The size of the dot indicates the GeneRatio (genes of interest in gene set, compared to total genes in category) and the color indicates the adjusted *p*-value (Fishers Exact Test, adjusted *p*-value < 0.05).

lines. The reproducibility of our findings underscores their reliability. While ideal models would include early embryonic or 8-cell-like cells, technical challenges limit biochemistry and interaction proteomics in these models. Our exploratory analysis identifies intriguing targets for future research, particularly for proteins transiently expressed during development. Although our results hint at interconnected mechanisms, further functional studies are essential to unravel the molecular dynamics of early germinal activation.

# Methods

## Reagents and tools

See Table 1.

## Cloning PRDL factors to pDONR-221

PRDL factors ARGFX, CPHX1, CPHX2, DPRX, DUXA, DUXB, TPRX1, and TPRX2 were first amplified in a two-step PCR reaction using our previously published clones (Madissoon et al, 2016) as templates, first using gene-specific primers and second using attB1/2 adapter primers (Dataset EV9) and cloned into a Gateway compatible entry clone using Gateway BP Clonase II (Invitrogen) according to manufacturer's instructions. Cloning primers and European Nucleotide Archive (ENA) accession numbers for template sequences are given in Dataset EV8. The entry clones were further cloned to Gateway compatible MAC-tag vectors. SATB1, SATB2, DPPA3, EGR2, ZNF263, KLF3, SATB1, SATB2, PITX1, PITX2, and GSC were obtained from the human Orfeome.

**Table 1.   Reagents and Tools.**

| Reagent/Resource | Reference or Source<br>*Source (public): Stock center, company, other labs.*<br>*Reference: list relevant study (e.g., Smith et al, 2017) if referring to previously published work; use "this study" if new. If neither applies: briefly explain.* | Identifier or Catalog Number<br>*Provide catalog numbers, stock numbers, database IDs or accession numbers, RRIDs or other relevant identifiers.* |
|---|---|---|
| **Experimental models** | | |
| *List cell lines, model organism strains, patient samples, isolated cell types etc. Indicate the species when appropriate.* | | |
| Human: HEK 293 cell line | ATCC | Cat# CRL-1573 |
| Human: HEK Flp-In T-REx 293 cell line | Thermo Fisher Scientific | Cat# R78007 |
| **Recombinant DNA** | | |
| *Indicate species for genes and proteins when appropriate* | | |
| MAC-GFP | Liu et al, 2018 | Addgene, plasmid no. 139636 |
| MAC-tag-N destination vector | Liu et al, 2018 | Addgene, plasmid no. 108078 |
| MAC-tag-C destination vector | Liu et al, 2018 | Addgene, plasmid no. 108077 |
| **Antibodies** | | |
| *Include the name of the antibody, the company (or lab) who supplied the antibody, the catalogue or clone number, the host species in which the antibody was raised and mention whether the antibody is monoclonal or polyclonal. Please indicate the concentrations used for different experimental procedures.* | | |
| Anti-HA tag antibody | Abcam | Cat # ab18181 |
| **Oligonucleotides and sequence-based reagents** | | |
| *For long lists of oligos or other sequences please refer to the relevant Table(s) or EV Table(s)* | | |
| PCR primers | This study | Dataset EV9 |
| **Chemicals, enzymes, and other reagents** | | |
| *e.g., drugs, peptides, recombinant proteins, dyes etc.* | | |
| Benzonase Nuclease | Santa Cruz | Cat# sc-202391 |
| Biotin | Pierce | Cat# 29129 |
| FuGENE 6 transfection reagent | Promega | Cat# E2691 |
| Hygromycin B | Thermo Fisher Scientific | Cat# 10687-010 |

**Table 1.** (continued)

| Reagent/Resource | Reference or Source<br><br>*Source (public): Stock center, company, other labs.<br>Reference: list relevant study (e.g., Smith et al, 2017) if referring to previously published work; use "this study" if new. If neither applies: briefly explain.* | Identifier or Catalog Number<br><br><br>*Provide catalog numbers, stock numbers, database IDs or accession numbers, RRIDs or other relevant identifiers.* |
|---|---|---|
| Mouse monoclonal Anti-HA—Agarose conjugated beads | Sigma-Aldrich | Cat# A2095 |
| Complete, Mini, EDTA-free Protease Inhibitor Cocktail | Roche | Cat#04693159001 |
| Dynabead protein G | Invitrogen | Cat#10004D |
| **Software** | | |
| *Include version where applicable* | | |
| CRAPome v1 | Mellacheruvu et al, 2013 | http://www.crapome.org/ |
| Cytoscape version 3.7 | Shannon et al, 2003 | http://www.cytoscape.org/ |
| MaxQuant version 1.6.4.3 | Cox and Mann, 2008 | http://www.biochem.mpg.de/5111795/maxquant |
| Progenesis LC-MS version 4.0 | Nonlinear Dynamics | http://www.nonlinear.com/progenesis/qi/ |
| Proteome Discoverer version 1.4 | Thermo Fisher Scientific | https://www.ThermoFisher.com/fi/en/home.html |
| SAINTexpress version 3.6 | Choi et al, 2011 | http://saint-apms.sourceforge.net/Main.html |
| Xcalibur version 2.7.0 | Thermo Fisher Scientific | https://www.ThermoFisher.com/fi/en/home.html |
| Fragpipe version 17 | Nesvizhskii Lab | https://fragpipe.nesvilab.org/ |
| FastQC v0.11.9 | Andrews et al, 2010 | https://www.bioinformatics.babraham.ac.uk/projects/fastqc/ |
| Trimmomatic v0.39 | Bolger et al, 2014 | http://www.usadellab.org/cms/?page=trimmomatic |
| Bowtie2 v2.4.4 | Langmead and Salzberg, 2012 | https://bowtie-bio.sourceforge.net/bowtie2/manual.shtml |
| MACS2 v2.2.9.1 | Y. Zhang et al, 2008 | https://github.com/macs3-project/MACS |
| UCSC tools | Kuhn et al, 2013 | https://genome.ucsc.edu/util.html |
| IGV v2.16.1 | Robinson et al, 2011 | https://igv.org/ |
| **Other** | | |
| *Kits, instrumentation, laboratory equipment, lab ware etc. that are critical for the experimental procedure and do not fit in any of the above categories can be listed here.* | | |
| Gateway LR Clonase Enzyme Mix | Life Technologies | Cat # 11791043 |
| Bio-Spin Chromatography Columns | Bio-Rad | Cat# 732-6008 |
| C18 reversed-phase spin columns | Nest Group | Cat# SEM SS18V |
| C18 macrospin columns | Nest Group | Cat# SMM SS18V |
| Q-Exactive mass spectrometer | Thermo Fisher Scientific | |
| EASY-nLC 1000 | Thermo Fisher Scientific | |
| NEBNext® Ultra™ II DNA Library Prep Kit for Illumina® | New England Biolabs (NEB) | Cat#E7103L |
| NEBNext Multiplex Oligos for Illumina | NEBNext Ultra II DNA Library Kit with Purification Beads | Cat#E7780S |
| Bio-Dot® Microfiltration System | Bio-Rad | 1703938 |

## MAC-tagged factors

Many of these factors have a homeodomain or homeodomain-like domain in the N-terminus, therefore C-terminal tags were used for most factors. For DPRX and CPHX2 the N-terminal tag produced higher number of identifications and was used as the final samples. The entry vector sequences and synthesis are from previous studies (Töhönen et al, 2015; Jouhilahti et al, 2016; Madissoon et al, 2016). The Gateway® recombination technology (Thermo Fisher Scientific) was utilized for creating all MAC-tagged constructs.

## Cell culture

To produce stable cell lines stably expressing MAC-tagged preimplantation factors, Flp-In T-REx 293 cells (Invitrogen, Life Technologies, R78007, cultured in manufacturer's recommended conditions) were co-transfected with the expression vector and the pOG44 vector (Invitrogen) using Fugene6 transfection reagent (Roche Applied Science). One day after transfection, cells were selected in 1% Streptomycin-Penicillin and 100 µg/ml Hygromycin for two weeks after which positive clones were

pooled and amplified. Green fluorescent protein (GFP) tagged with MAC-tag was used as a negative control and processed parallel to the bait proteins.

Stable cell line was expanded to 80% confluence in $20 \times 150$ mm cell culture plates. Ten plates were used for AP-MS, in which 2 µg/ml tetracycline was added for 24 h induction, and ten plates for BioID, in which 50 µM biotin in addition to tetracycline, was added for 24 h before harvesting. An exception to this was DUX4, which was cultivated for 20 h, respectively, due to DUX4 induction killing a large number of cells. Cells from five fully confluent dishes were pelleted as one biological sample. In total two biological replicates in two different approaches were produced. Samples were snap frozen and stored at –80 °C.

## Affinity purification

In the AP-MS sample purification the sample was lysed in 3 ml ice-cold Lysis Buffer I (1% n-Dodecyl beta-D-maltoside, 50 mM Hepes, pH 8.0, 150 mM NaCl, 50 mM NaF, 1.5 mM NaVO₃, 5 mM EDTA, 0.5 mM PMSF and Sigma Proteinase Inhibitor). In the BioID sample the cell sample was lysed in 3 ml ice-cold Lysis Buffer I, supplemented with 1 µl Benzonase per sample and sonicated. Lysed samples were centrifuged at 16,000x for 15 min, and again 10 min to produce cleared lysate, that was loaded on Bio-Rad spin columns that had 400 µl Strep-Tactin beads (IBA, GmbH) prewashed with Lysis Buffer I. The loaded beads were washed 3× 1 ml with Lysis Buffer I, and 4 × 1 ml with Wash Buffer (50 mM tris-Hcl, pH 8.0, 150 mM NaCl, 50 mM NaF, 5 mM EDTA). To eluate sample, the beads were resuspended in 2 × 300 µl Elution Buffer (50 mM Tris-Hcl, pH 8.0, 150 mM NaCl, 50 mM NaF, 5 mM EDTA, 0.5 mM Biotin) for 5 min and eluates were collected into an Eppendorf tube, followed by a reduction of the cysteine bonds with 5 mM Tris(2-carboxyethyl)phosphine (TCEP) for 30 min at 37 °C and alkylation with 10 mM iodoacetamide. The proteins were then digested to peptides with sequencing grade modified trypsin (Promega, V5113) at 37 °C overnight. Samples were then desalted by C18 reversed-phase spin columns according to manufacturer's instructions. The sample was dried in a vacuum centrifuge and reconstituted to a final volume of 30 µl in 0.1% TFA and 1% Acetonitrile.

## Liquid chromatography-mass spectrometry (LC-MS)

Analysis was performed on a Q-Exactive mass spectrometer with an EASY-nLC 1000 Liquid Chromatograph Q Exactive™ Hybrid Quadrupole-Orbitrap™ system via an electrospray ionization sprayer (Thermo Fisher Scientific), using Xcalibur version 3.0.63. Peptides were eluted from the sample with a C18 precolumn (Acclaim PepMap 100, 75 µm × 2 cm, 3 µm, 100 Å; Thermo Scientific) and analytical column (Acclaim PepMap RSLC, 65 µm × 15 cm, 2 µm, 100 Å; Thermo Scientific), using a 60 min buffer gradient ranging from 5 to 35% Buffer B, then a 5 min gradient from 35 to 80% Buffer B and 10 min gradient from 80 to 100% Buffer B (0.1% formic acid in 98% acetonitrile and 2% HPLC grade water). 4 µl of peptide samples was loaded to a cooled autosampler. Data-dependent FTMS acquisition was in positive ion mode for 80 min. A full scan (200–2000 $m/z$) was performed with a resolution of 70,000 followed by top10 CID-MS² ion trap scans with a resolution of 17,500. Dynamic exclusion was set for 30 s. Database search was performed with MSFragger (v17) on the Reviewed human proteome in

UniProtKB/SwissProt databases (http://www.uniprot.org, downloaded Nov. 2021). Trypsin was selected as the cleavage enzyme and maximum of 2 missed cleavages were permitted, precursor mass tolerance at ±15 ppm and fragment mass tolerance at 0.05 Da. Carbamidomethylation of cysteine was defined as a static modification. Oxidation of methionine and, for BioID samples, biotinylation of lysine and N-termini were set as variable modifications. All reported data were based on high-confidence peptides assigned in MSFragger (FDR < 0.01).

## Identification of statistical confidence of interactions

Significance Analysis of INTeractome (SAINT)-express version 3.6.3 and Contaminant Repository for Affinity Purification (CRAPome, http://www.crapome.org) were used to discover statistically significant interactions from the AP-MS data. The preimplantation factor LC-MS data was run alongside a large dataset of other transcription factors, as well as a large GFP control set. Final results represent proteins with a BFDR score lower than 0.01, and in less than 10% of CRAPome database experiments except in cases where AvgSpec is three times higher than AvgSpec in CRAPome experiments. For BioID we used a contaminant database we constructed ourselves (Dataset EV2).

For AP-MS and BioID we used stringent SAINTexpress BFDR < 0.01 cut off and compared our results to CRAPome and BioID-MS contaminant libraries that we constructed ourselves. We allowed only <10% of control samples to have detected our preys, however, allowing those where our Average Spectral count was three times higher than Average Spectral count in our filtering set. In BioID-MS filtering we eliminated preys that were tagged contaminants in either All or NLS GFPs. After significant filtering, total 1 BioID-MS 1389 and 500 AP-MS high-confidence interactions were identified for the 21 baits excluding bait detection (Datasets EV3 and EV4).

## Construction of localization targeted GFP database

The database was constructed from 156 MAC-tagged GFP replicates, using the same MAC-tag purification pipeline that we used for the actual samples (Liu et al, 2018; Liu et al, 2020). The peptide and corresponding protein identification from the MS raw files were performed using the MSFragger pipeline. In this BioID contaminant database, we eliminated runs with less than 1000 detected proteins from further consideration in our copurification database to avoid weak runs affecting the final database values. After stringent filtering a total of 113 GFP runs were used to construct the copurificant database (Dataset EV2). The baits in these runs consisted of GFPs with no specialized localization tags (64 in total), MYR-membrane localization tag (24 total), NES18 (4), and NLS-localization tag (26). In total, we identified 4014 proteins through GFP BioID samples. We next used similar filtering options as presented in the widely used CRAPome database (Mellacheruvu et al, 2013), total percentage of experiments where prey was detected, and average and maximum spectral count of prey if detected. Further we split the values into more detailed columns, combining all GFP runs, only NLS localized GFPs, and only MYR localized GFPs. 217 proteins were identified only in our set compared to CRAPome, and conversely 4419 reviewed UniProt proteins were identified in CRAPome and not our set.

We also created an AP-MS contaminant list using 52 GFP samples, which were filtered down to those with at least 300 preys identified: 12 NLS GFPs, 1 NS18 GFP, and 20 generic GFPs (total 33). We identified in total 1557 proteins through AP-MS GFP samples. Due to the low number of GFPs, identified preys, and low number of samples, we chose to use CRAPome to filter AP-MS samples for contaminants and copurificants (CRAPome has 716 negative control runs in the database offering far better filtering coverage).

## Co-immunoprecipitation

To validate the interaction pairs by Co-IP, HEK293 cells ($5 \times 10^5$ per well) in 6-well plate were co-transfected with Strep-HA-tagged (500 ng) bait and V5-tagged (500 ng) prey constructs using Fugene 6 transfection reagent (Promega). After 24 h of transfection, cells were rinsed with ice-cold PBS and lysed with 1 ml HENN lysis buffer per well (50 mM HEPES pH 8.0, 5 mM EDTA, 150 mM NaCl, 50 mM NaF, 0.5% IGEPAL, 1 mM DTT, 1 mM PMSF, 1.5 mM $Na_3VO_4$, 1 × Protease inhibitor cocktail) on ice. The cell lysate was collected, and a clear supernatant was obtained by centrifugation (16,000 × $g$, 20 min, 4 °C). 30 µl of Strep-Tactin® Sepharose® resin (50% suspension, IBA Lifesciences GmbH) was washed in a microcentrifuge tube twice with 200 µl HENN lysis buffer (4000 × $g$, 1 min, 4 °C). The clear lysate was added to the pre-washed Strep-Tactin beads and incubated for 1 h on a rotation wheel at 4 °C. After incubation, the beads were collected by centrifugation and washed three times with 1 ml HENN lysis buffer (4000 × $g$, 30 s, 4 °C). After the last wash, 60 µl of 2 × Laemmli sample buffer (Bio-Rad, 1610737) was added directly to the beads and boiled at 95 °C for 5 min. Samples were later used for immunodetection via dot-blot.

## Dot-blot

The Bio-Dot® Microfiltration System (Bio-Rad, 1703938) was assembled according to the manufacturer's instructions. The nitrocellulose membrane was pre-washed with TBS to hydrate the membrane. Ten microliters of sample was spotted onto the nitrocellulose membrane in the center of the well and drained under vacuum pressure. Non-specific sites were blocked with 5% fat-free milk in TBS-T (0.05% Tween-20 in TBS) for 60 min at RT with gentle shaking. The membrane was then incubated with primary antibody in TBS-T (mouse anti-V5 with a 1:5000 dilution) overnight at 4 °C. The membrane was washed three times for 10 min with TBS-T followed by incubation with secondary antibody conjugated with HRP (goat anti-mouse IgG conjugated with horseradish peroxidase with a 1:2000 dilution) for 60 min at RT with gentle shaking. The membrane was washed three times for 10 min with TBS-T followed by one additional wash with TBS on a shaker. Amersham™ ECL™ Prime (Cytiva) solution was added to the membrane and incubated for 5 min prior to imaging the blot using iBright Imaging Systems (Thermo Fisher). The same membrane was then stripped by incubating with Restore Plus Stripping buffer (Thermo Fisher) for 15 min and was re-blocked with 5% fat-free milk in TBS-T for 60 min at RT with gentle shaking. The membrane was then incubated with the other primary antibody in TBS-T (mouse anti-HA with a 1:2000 dilution) overnight at 4 °C for different detections.

## Validation of BioID-MS interactions in iPSCs

Human induced pluripotent stem cell line HEL24.3 (Trokovic et al, 2015) was cultured on recombinant human laminin 521 (LN521, Biolamina) in Essential 8 culture medium (Gibco) and passaged with EDTA (Versene, Gibco). For the generation of the doxycycline inducible MAC3 tag fusion factor cells, the cells were electroporated with Neon transfection system using 1100 V, 20 ms and 2 × pulse settings with PiggyBac plasmids containing the transposase, rtTA and the MAC3 tag fusion factor. Electroporated cells were grown overnight in 10 µM ROCK inhibitor (Y-27632 (hydrochloride), 10005583, Cayman Chemicals) containing medium after which the cells were selected with 0.5 µg/ml Puromycin (J67236.XF, Fischer Scientific) and 250 µg/ml G418 (11811023, Gibco) for 6 days. For the induction of the MAC3 tag fusion factors the cells were treated with 2 µg/ml Doxycycline (BP2653-1, Fischer Scientific) overnight after which the cells were treated with 50 µM Biotin and 2 µg/ml Doxycycline for 3 h. Samples were collected with ice-cold EDTA and snap frozen.

Cells were lysed in ice-cold lysis buffer containing 0.5% IGEPAL, 50 mM HEPES (pH 8.0), 150 mM NaCl, 50 mM NaF, 1.5 mM NaVO₃, 5 mM EDTA, 0.5 mM PMSF, and protease inhibitors (Sigma-Aldrich)). Cleared lysate was obtained by centrifugation. The lysate was then mixed with prewashed Strep-Tactin® Sepharose® resin (IBA) and subjected for rotating for 2 h. The biotin-containing buffer (50 mM Tris-Hcl, pH 8.0, 150 mM NaCl, 50 mM NaF, 5 mM EDTA, 0.5 mM Biotin) was used to elute the pure protein complexes from beads. For the subsequent study, the beads were resuspended in buffer (50 mM Tris-Hcl, pH 8.0, 150 mM NaCl, 50 mM NaF, 5 mM EDTA). For MS analysis, both elution and beads samples were reduced, alkylated, and digested to peptides. Finally, the peptide samples were desalted using Nest Group C18 Macrospin columns. The desalted samples were examined with an Evosep One liquid chromatography system connected to a Bruker TimsTOF Pro hybrid trapped ion mobility quadrupole TOF mass spectrometer via a CaptiveSpray nano-electrospray ion source. For peptide separation using the 60 samples per day approaches (21 min gradient time), an 8 cm 150 m column with 1.5 m C18 beads (EV1109, Evosep) was utilized. The concentrations of formic acid in mobile phases A and B were 0.1% in water and 0.1% in acetonitrile, respectively. The MS analysis was carried out in the positive-ion mode utilizing data-dependent acquisition (DDA) in PASEF mode, with a total of 10 PASEF scans each topN acquisition cycle.

## Chromatin immunoprecipation sequencing (ChIP-seq)

HEK293 cells were seeded in 10 cm dishes; 24 h later, tetracycline at a final concentration of 2 µg/ml was added to induce the expression of targeted protein. ChIP-seq assays performed as previously described (Ma et al, 2022). In brief, HEK293 cells were cross-linked with 1% formaldehyde for 10 min at room temperature. The reaction was quenched with 125 mM glycine. Cells were collected after washing twice with precooled PBS and resuspended in hypotonic lysis buffer (20 mM Tris-Cl, pH 8.0, with 10% glycerol, 10 mM KCl, 2 mM DTT, and complete protease inhibitor cocktail (Roche)) to isolate nuclei. The nuclear pellets were washed with precooled PBS and resuspended in a 1:1 ratio of SDS lysis buffer (50 mM Tris-HCl, pH 8.1, with 1% SDS, 10 mM EDTA, and complete protease inhibitor) and ChIP dilution buffer (16.7 mM Tris-HCl, pH 8.1, with 0.01% SDS, 1.1% Triton X-100, 1.2 mM EDTA, 167 mM NaCl and complete protease inhibitor).

A Q800R sonicator (QSonica) was used to generate an average size of 300 bp chromatin fragments at 4 °C. Dynabead protein G (Invitrogen) was washed with blocking buffer (0.5% BSA in IP buffer (20 mM Tris-HCl, pH 8.0, with 2 mM EDTA, 150 mM NaCl, 1% Triton X-100, and protease inhibitor cocktail)) and incubated with an antibody against HA (ab18181, Abcam). Chromatin lysate was precipitated with Dynabead protein G for 12 h and then washed 4 times with washing buffer (50 mM HEPES, pH 7.6, 1 mM EDTA, 0.7% sodium deoxycholate, 1% NP-40, 0.5 M LiCl) and 2 times 100 mM ammonium hydrogen carbonate (AMBIC) solution. The DNA-protein complexes extracted from the beads by elution in extraction buffer (10 mM Tris-HCl, pH 8.0, with 1 mM EDTA and 1% SDS), proteinase K and NaCl were added to reverse the cross-links. A Mini-Elute PCR purification kit (Qiagen) was used to purify the DNA. The purified DNA was subjected to ChIP-seq library preparation using the NEBNext® Ultra™ II DNA Library Prep Kit (E7103L, NEB). The DNA libraries were sequenced by a NovaSeq 6000 S4 sequencing system (Illumina, San Diego, CA, USA).

## Bioinformatics analysis of ChIP-seq data

The ChIP-seq library was sequenced and generated single-end reads of 150 bp. FastQC was applied to investigate the quality of raw sequence reads. Trimmomatic (Bolger et al, 2014) was employed for the quality control with following parameters: TruSeq3-SE.fa:2:30:10 SLIDING-WINDOW:5:20. Read lengths less than 10 bp were filtered out. Bowtie2 (Langmead and Salzberg, 2012) was subsequently used to map the processed reads to the human genome hg38 with default settings. MACS2 (Zhang et al, 2008) was applied for peak calling with default settings. UCSC tools were applied to generate ChIP-seq coverage signals and Integrative Genomics Viewer (IGV) (Robinson et al, 2011) was used for visualization.

## Overrepresentation analysis

Overrepresentation analysis of statistically significant interactions in each Bait interactome was done using enrichR R-package (Chen et al, 2013; Kuleshov et al, 2016; Xie et al, 2021). We performed GO, Reactome, CORUM, KEGG, and domain enrichment analysis. Enrichment cut-off was set FDR < 0.05. Enrichment analysis figures were plotted using the R-package ggplot2 (https://ggplot2.tidyverse.org).

## Clustering analysis

Clustering was done using tSNE (R-package tsne), Ward's hierarchical clustering (using Eucledian distance), and Correlation Analysis were done with Prohits-viz (https://prohits-viz.org/) (Knight et al, 2015).

## MS-microscopy analysis

Localization of baits was studied using MS Microcopy (http://proteomics.fi/) (Liu et al, 2018).

## Interaction network

Protein interaction networks were constructed from filtered BioID-MS data that was imported to Cytoscape 3.6.0.

## Multiple sequence alignment of amino acid sequences

Amino acid sequences for TPRX2, CPHX1, and CPHX2 were converted from Töhönen et al (2015), Jouhilahti et al (2016) vector sequences with ORFfinder. Sequences for human ARGFX, LEUTX, CRX, DPRX, OTX1, OTX2, and mouse CRX and Drosophila melanogaster HMOC were fetched from UniProt (peptide sequence to FASTA). Alignment of amino acid sequences was done using MAFFT using MAFFT version 7 at (https://mafft.cbrc.jp/alignment/server/) with default settings (MAFFT-L-INS-i) with BLOSUM62 scoring. UPGMA tree was constructed with JTT substitution based on the MSA alignment and visualized with Phylo.io.

## Annotation of ChIPSeq peaks

Annotation plots for genomic regions were done with ChIPSeeker R-package (Yu et al, 2015), with promoter regions defined as 3000 kb up or downstream from known GENCODE TSS sites. Plotting of genomic regions was done using Integrative Genomics Viewer (Robinson et al, 2011).

## Motif finding: MEME suite

To analyze which motifs were found in the genomic coordinates we had we used MEMESuite (Bailey et al, 2015). MEME (Bailey and Elkan, 1994) for all genomic data was run with settings mode:"anr", nmotifs = 25, min width = 6, maxwidth = 50, minimum sites 50, csites = 3000, time = 30000. Further, we analyzed what motifs were enriched in each data set with the MemeStuite tool SpaMo. SpaMo was run with default settings using the motif database HOCO-MOCO core human version v11. Known motifs for the studies proteins were searched for and downloaded from JASPAR database (https://jaspar.genereg.net/) (Castro-Mondragon et al, 2022).

## Comparison of ChIP-Seq peaks to ENCODE data

We downloaded data for KLF4 (ENCSR265WJC), ZNF263 (ENCSR000EVD), and EGR2 (ENCSR919CZU) from the ENCODE Portal (Luo et al, 2020) (downloaded 5.2022, https://www.encodeproject.org/). In each case, we used the "IDR ranked peaks" bed narrowPeak files. We compared these bed files to the bed files produced from our analysis using bedtools closest. Overlapping peaks are identified with a distance of 0.

## Comparison of ChIP-Seq peaks with embryonic ATAC-seq study by Wu et al, 2018

Data from GSE101571 (Wu et al, 2018) was downloaded through NCBI data repository (accessed 21.3.2021), we used bigwig and bed files from the study. Further, wig files for 4 cell, 8 cell, and ICM stages and primed hESCs from the same study were converted to tdf format and into vector graphics using Integrative Genomics Viewer (Robinson et al, 2011). (Kubitz et al, 2022; Rezsohazy et al, 2015; Yu et al, 2022).

# Data availability

The raw and processed ChIP-seq data has been upload to the European Nucleotide Archive (ENA) (https://www.ebi.ac.uk/ena/browser/home)

with accession PRJEB58808 (https://www.ebi.ac.uk/ena/browser/view/PRJEB58808). The raw mass spectrometry files have been uploaded to MassIVE (MSV000091321) (ftp://massive.ucsd.edu/v01/MSV000091321).

## Peer review information

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

## Acknowledgements

We thank Dr. Shintaro Katayama and Dr. Masahito Yoshihara for their contributions in identifying the EGA genes and their transcriptomics functions. We thank Sini Miettinen (Proteomics Unit, Institute of Biotechnology & HiLIFE). This work was supported by the Academy of Finland (#288475 and CoVIDD #336470 for MV), Academy of Finland (#336470), Sigrid Jusélius Foundation, Jane and Aatos Erkko Foundation, Finska Läkaresällskapet, and Finnish Cultural Foundation for funding. The authors wish to also acknowledge CSC – IT Center for Science, Finland for computational resources.

## Author contributions

**Lisa Gawriyski:** Conceptualization; Data curation; Software; Formal analysis; Funding acquisition; Validation; Investigation; Visualization; Methodology; Writing—original draft; Project administration; Writing—review and editing. **Zenglai Tan:** Data curation; Software; Formal analysis; Validation; Investigation; Visualization; Methodology; Writing—review and editing. **Xiaonan Liu:** Data curation; Software; Formal analysis; Validation; Investigation; Visualization; Methodology; Writing—review and editing. **Iftekhar Chowdhury:** Formal analysis; Investigation; Methodology; Writing—review and editing. **Dicle Malaymar-Pinar:** Formal analysis; Investigation; Methodology; Writing—review and editing. **Qin Zhang:** Software; Formal analysis; Investigation; Methodology; Writing—review and editing. **Jere Weltner:** Formal analysis; Investigation; Methodology; Writing—review and editing. **Eeva-Mari Jouhilahti:** Resources; Methodology; Writing—review and editing. **Gong-Hong Wei:** Resources; Supervision; Funding acquisition; Validation; Methodology; Project administration; Writing—review and editing. **Juha Kere:** Conceptualization; Resources; Data curation; Supervision; Funding acquisition; Validation; Project administration; Writing—review and editing. **Markku Varjosalo:** Conceptualization; Resources; Data curation; Formal analysis; Supervision; Funding acquisition; Validation; Visualization; Methodology; Writing—original draft; Project administration; Writing—review and editing.

## Disclosure and competing interests statement

The authors declare no competing interests.

# Expanded View Figures

**Figure EV1.  Additional info and methods used in this study.**

(**A**) Expression of other baits in the bait set in the embryonic transcriptomics dataset Yan et al (2013). (**B**) Overview of the methods used in this study. We first produced a stable cell line containing the protein of interest (bait), and then performed affinity purification mass spectrometry using two methods and chromatin immunoprecipitation sequencing in the same HEK293 cell line. We combined the produced data and performed bioinformatic analysis. (**C**) Domain prediction for CPHX1 and CPHX2. (**D**) Homeodomain alignment of all homeodomain proteins in the dataset.

**A**

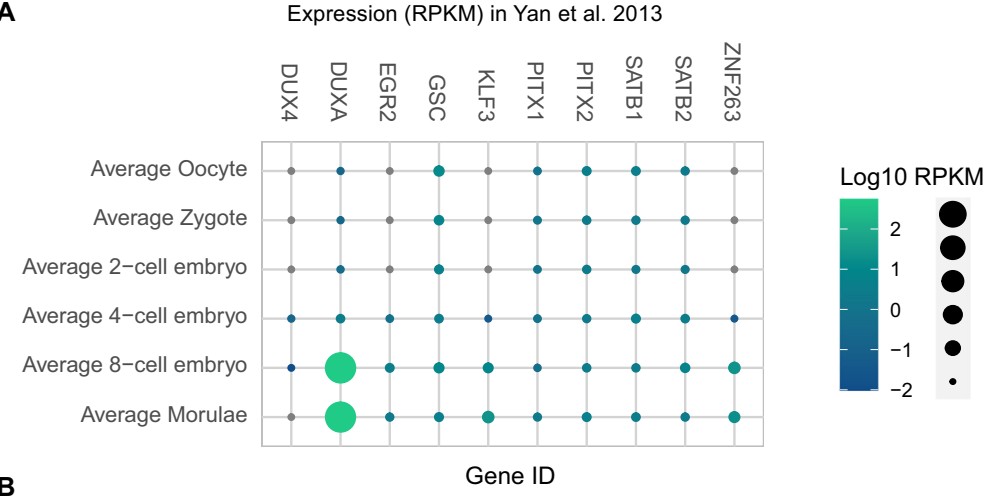

**B**

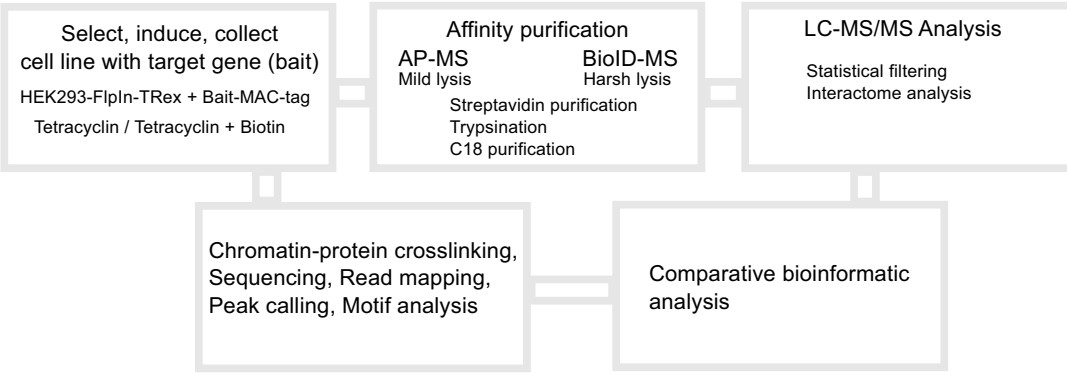

**C**

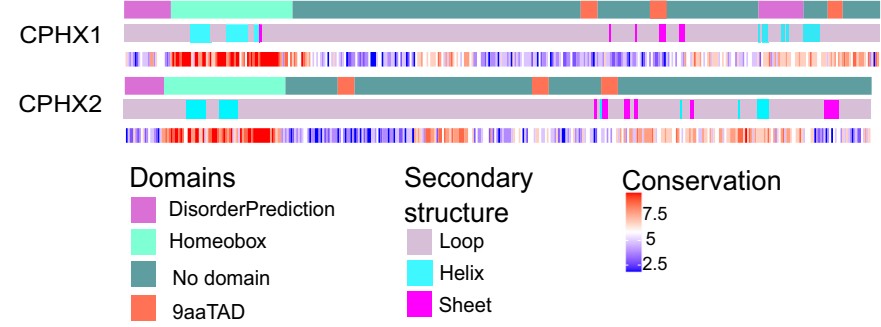

**D**

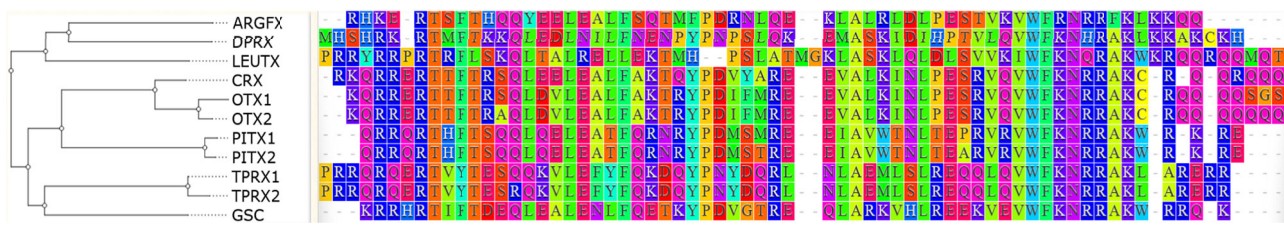

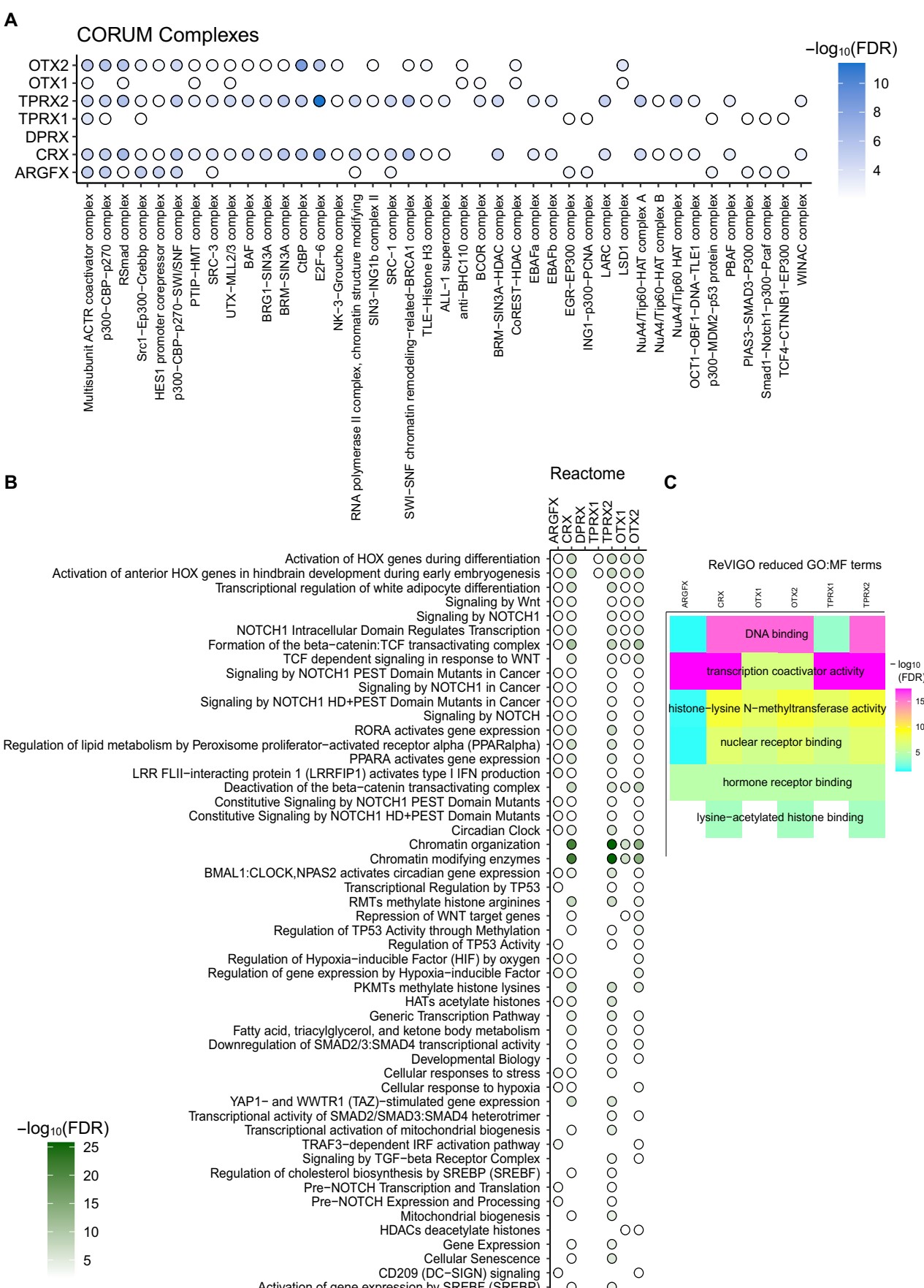

**Figure EV2. Enrichment analysis of PRDL bait interactomes.**

(A) CORUM complex enrichment analysis for PRDL bait interactomes, color indicates inverse significant enrichment of complex ($-\log_{10}$ (FDR)). Cutoff for image FDR < 0.05, Fishers Exact Test. (B) Reactome pathway enrichment analysis for PRDL bait interactomes, color indicates inverse significant enrichment of pathways ($-\log_{10}$ (FDR)). Cutoff for image FDR < 0.05´, Fishers Exact Test. (C) Gene ontology molecular function enrichment analysis of PRDL bait interactomes, color indicates inverse significant enrichment of pathways ($-\log_{10}$ (FDR)). Cutoff for image FDR < 0.05, Fishers Exact Test.

**A**

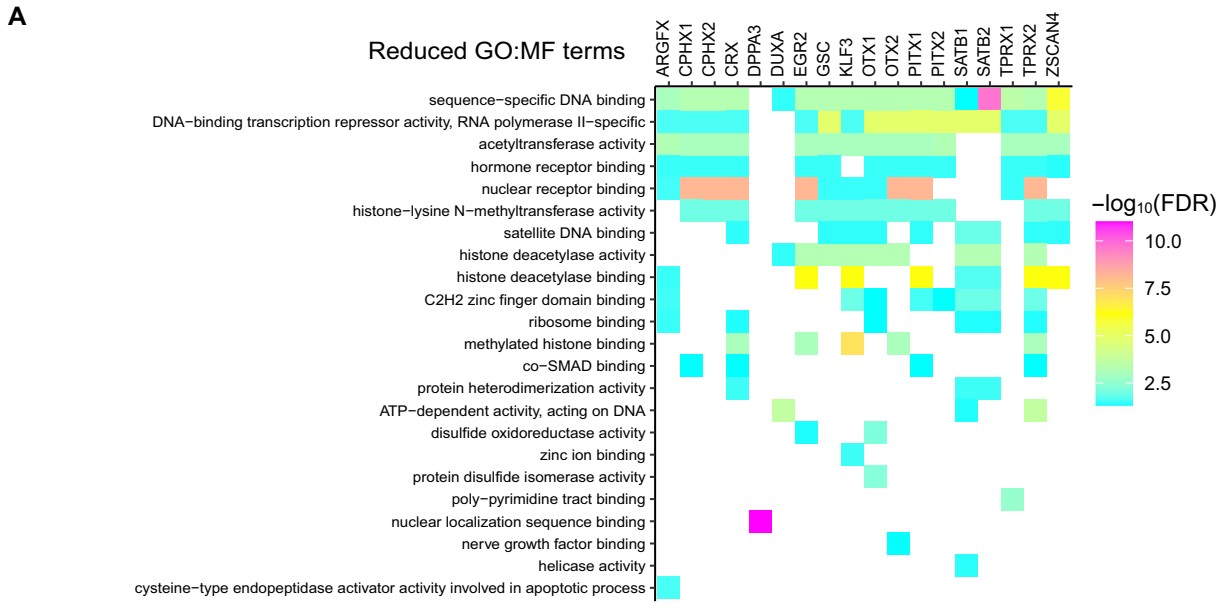

**B**

**Figure EV3. Gene ontology enrichment analysis of BioID-MS HCIs.**

(A) Gene ontology molecular function enrichment analysis of all BioID-MS HCIs, color indicates inverse significant enrichment of pathways ($-\log_{10}$ (FDR)). Cutoff for image FDR < 0.05, Fishers Exact Test. (B) Gene ontology biological process enrichment analysis of all BioID-MS HCIs, color indicates inverse significant enrichment of pathways ($-\log_{10}$ (FDR)). Cutoff for image FDR < 0.05, Fishers Exact Test.

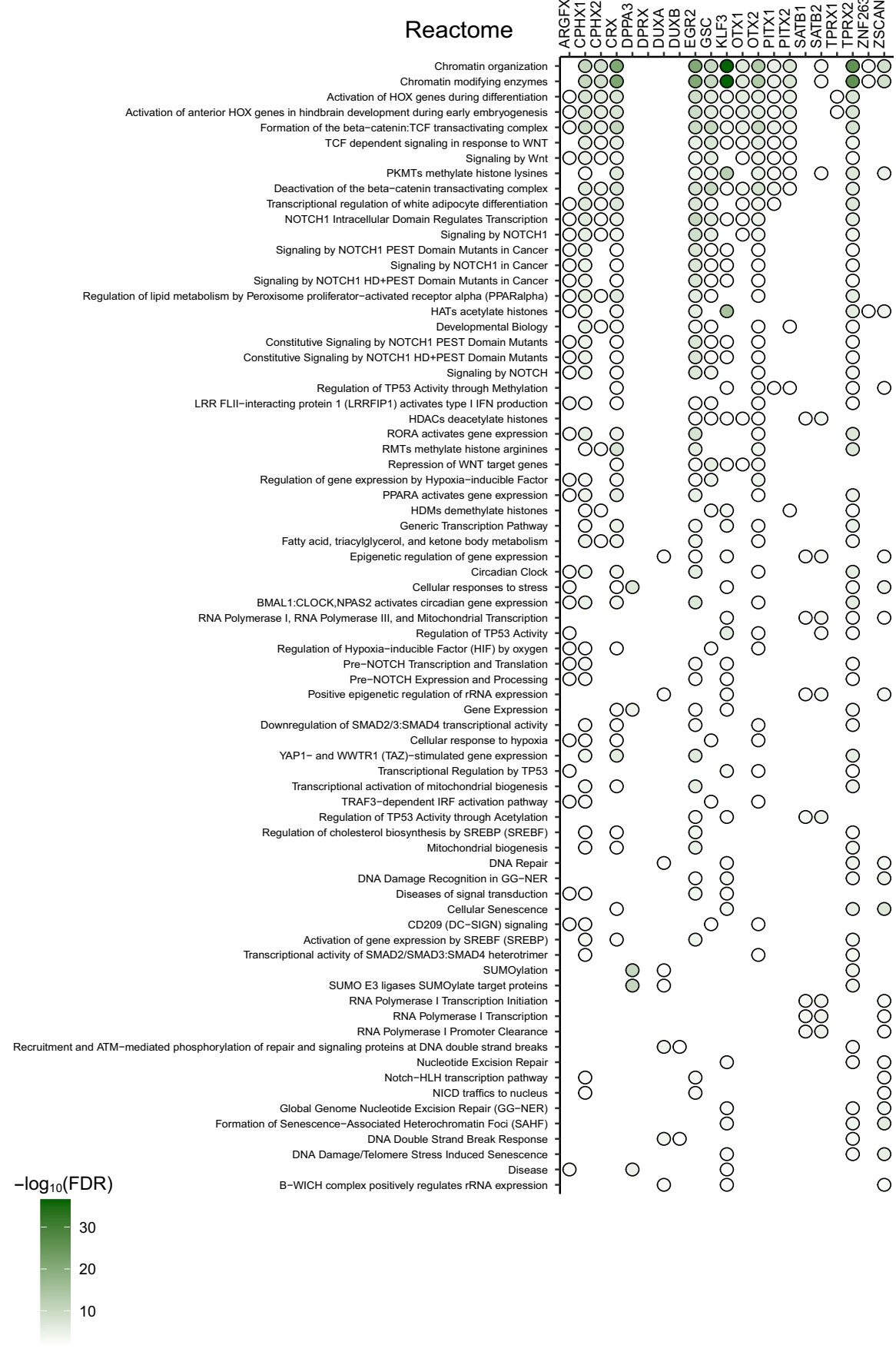

**Figure EV4. Reactome pathway and ChIP-seq peak overlap analysis.**

Reactome pathway enrichment analysis of all bait interactomes, color indicates inverse significant enrichment of pathways ($-\log_{10}$ (FDR)). Cutoff for image FDR < 0.05, Fishers Exact Test.

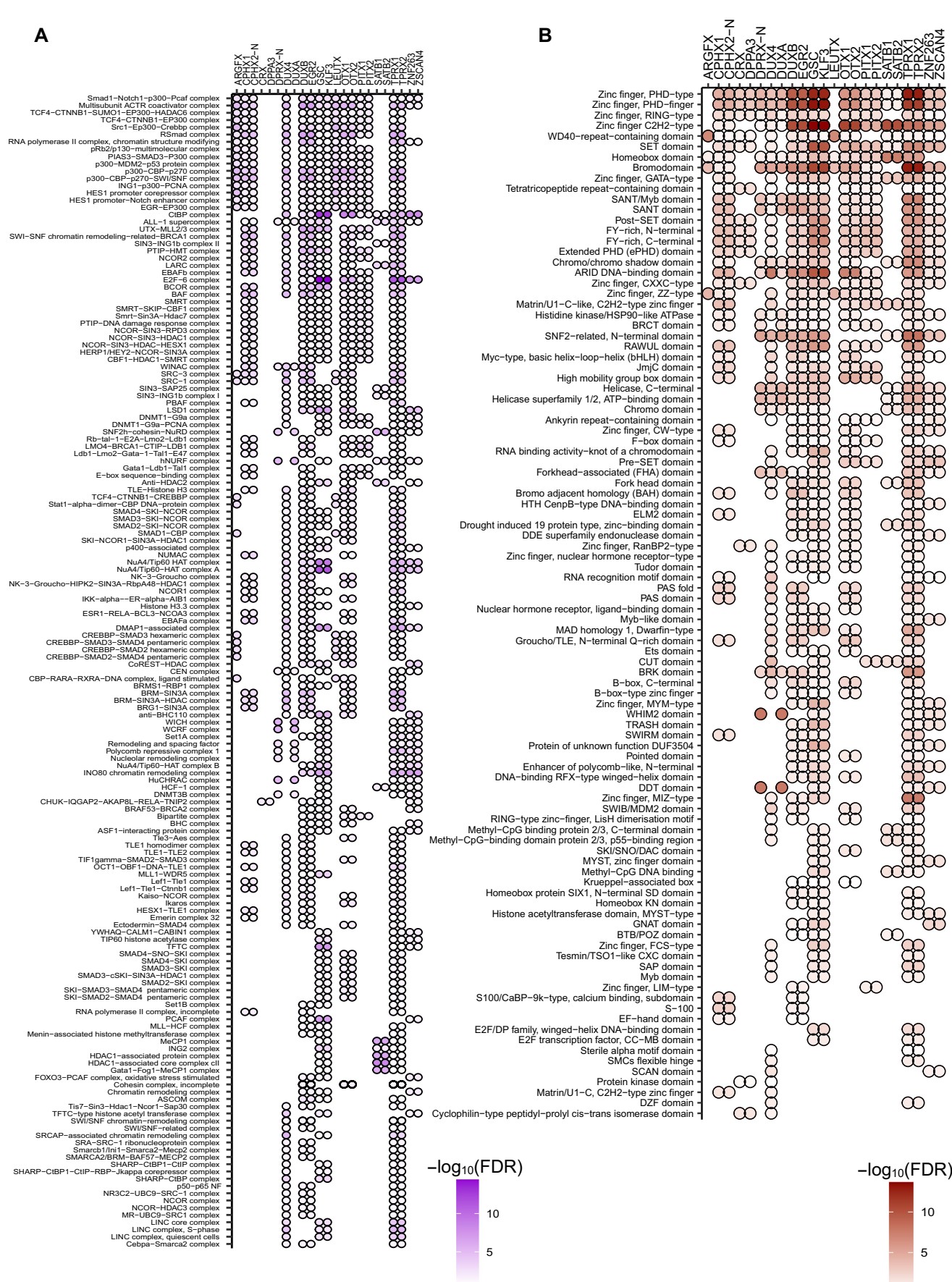

**Figure EV5. Enrichment analysis of expanded BioID-MS dataset.**

(A) Significantly enriched (Fishers Exact Test, FDR < 0.01) protein complexes in the interactomes of each bait. Protein complexes were obtained from the CORUM database. Only complexes enriched in more than two baits are drawn. Increased blue color indicates inverse statistical significance ($-\log_{10}$ (FDR)). Complexes are ordered by frequency. (B) Statistically significant enriched InterPro domains (FDR < 0.01) in the interactome of each bait, indicated by the number of shared baits. Color indicates $-\log_{10}$ (FDR) of enrichment analysis. Domains are ordered by $-\log_{10}$ (FDR) and frequency.

**A**

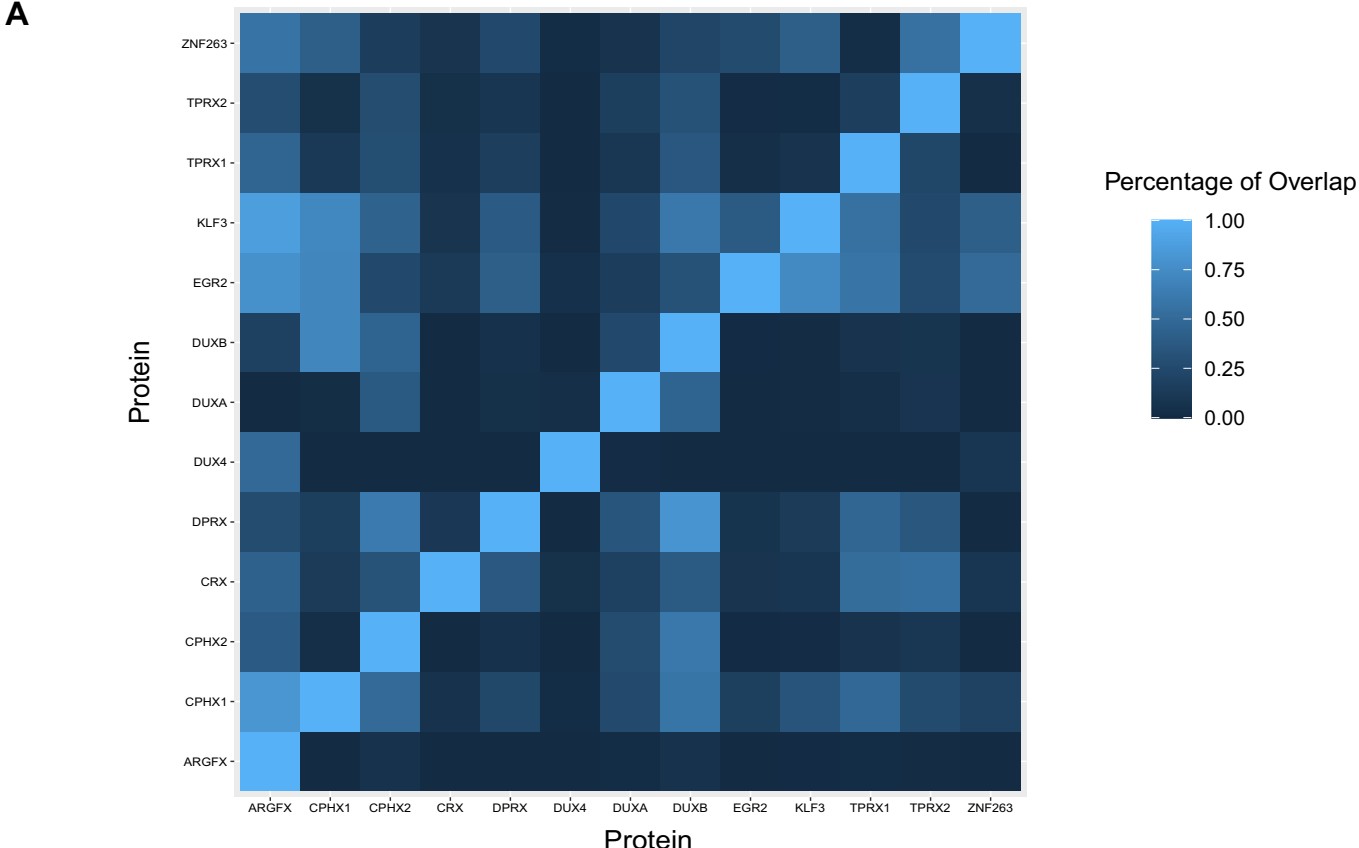

**B**

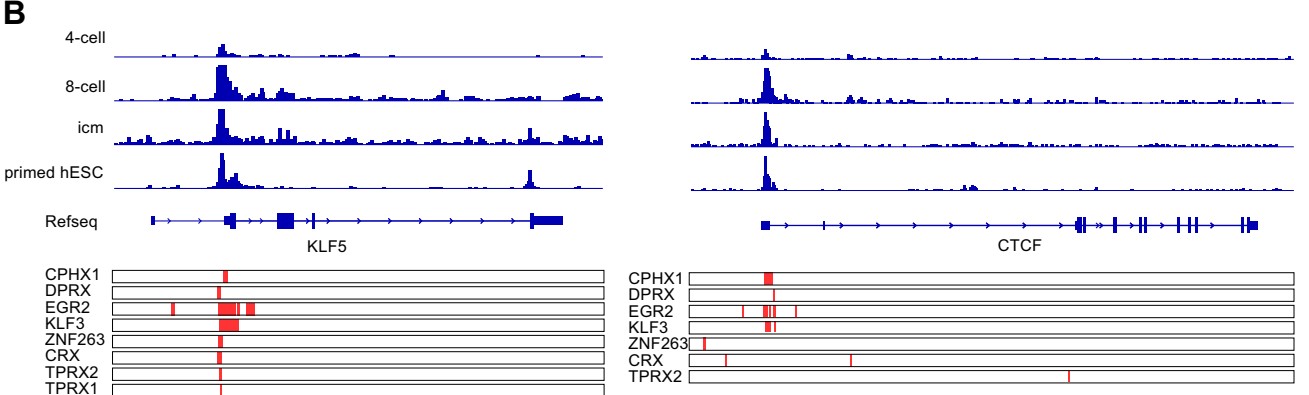

**Figure EV6.  ChIP-Seq Peaks over key EGA genes.**

(**A**) Percentage overlap of identified ChIP-seq peaks. Overlaps were identified using bedtools closest and are shown as percentages from 0 to 1. (**B**) The embryonic ATAC-seq data from Wu et al (2018) from 4-cell, 8-cell, and ICM embryos and primed hESCs over key gene regions identified through our datasets. Genomic locations of differential ChIP-seq peaks are shown in red, relative to their position to key genes (RefSeq annotation).

