## [Peer Review File · EMBO Reports]

Interaction network of human early embryonic transcription factors

Lisa Gawryski, Zenglai Tan, Xiaonan Liu, Iftekhar Chowdhury, Dicle Malaymar Pinar, Qin Zhang, Jere Weltner, Eeva-Mari Jouhilahti, Gong-Hong Wei, Juha Kere, and Markku Varjosalo

Corresponding author(s): Markku Varjosalo (markku.varjosalo@helsinki.fi)

Review Timeline:

Transfer Date:	14th Dec 23
Editorial Decision:	2nd Jan 24
Revision Received:	12th Jan 24
Accepted:	18th Jan 24

Editor: Esther Schnapp

Transaction Report: A revised version of this manuscript was transferred to EMBO reports following peer review at The EMBO Journal.

Date: 10th Apr 23 16:27:08

Last Sent: 10th Apr 23 16:27:08

From: k.anderson@embojournal.org

To: markku.varjosalo@helsinki.fi

Subject: Manuscript EMBOJ-2023-113907 - Decision

Message: Dear Prof. Varjosalo,

Thank you for submitting your manuscript for consideration by the EMBO Journal. It has now been seen by two referees whose comments are shown below.

Given these opinions and the fact that the EMBO Journal can only afford to accept papers which receive enthusiastic support from a majority of referees, I am afraid we cannot offer to publish it here. In this case, both referees are concerned that the cell line used to generate the data are not the most appropriate to the hypothesis and we feel that addressing this would require significantly more time than is typically allowed. However, we recognize the interest the findings and would therefore be open to reconsider a revised version of the study if you are able to fully resolve the main concerns of the referees in the future. Please however note that in such a case, we will again assess novelty and advance at the timepoint of submission and may involve alternative or additional referees if needed. In case you have any questions regarding this procedure, please feel free to contact me.

That said, if you do not wish to carry out a full revision, we encourage you to transfer your study to our not-for-profit open-access sister journal, Life Science Alliance (LSA). We shared your manuscript and the accompanying reviews with LSA Executive Editor, Eric Sawey, who is interested in these findings, and would like to publish this manuscript at LSA pending the following revisions: - Address Reviewer 1's Specific Comments. - Address Reviewer 2's comments for the paper to be considered as a Resource. We encourage you to use the link below to transfer your manuscript to LSA. You do not need to revise the manuscript before transferring it to LSA. Once you transfer, Dr. Sawey will email you an invitation to revise and resubmit, listing the same revision requests as mentioned above. Please feel free to reach out at e.sawey@life-science-alliance.org if you have any questions about the LSA journal, the transfer process or the revisions requested.

Thank you in any case for the opportunity to consider this manuscript. I am sorry we cannot be more positive on this occasion, but we hope nevertheless that you will find our referees' comments helpful.

Yours sincerely,

Kelly M Anderson, PhD
Editor
The EMBO Journal
k.anderson@embojournal.org

Referee #1:

Gawriyski et al.

In this study Gawriyski et al. use a combination of structure prediction, proteomics methodology, CHIP-seq and computational analyses to define the transcription factor networks that are likely to operate during embryonic genome activation (EGA). The authors focused on the ETCHbox family because this group of TF had been previously implicated in EGA. Structural predictions identified DNA binding domains and transactivation domains for a number of family members. The authors then used affinity purification mass spectrometry and proximity dependent biotin identification to discover protein protein interactions for a subset of the genes of interest. Thereafter more interactions were identified using a wider set of developmentally relevant TF in the interactome analysis. Ontology and expression studies further supported the likely role of high confidence interactions identified in transcriptional regulation (transcription factors or chromatin modifiers) during early development. CHIP-seq studies revealed that many factors interacted with binding sites of genes active during the eight cell stage of development (a major wave of EGA).

With the availability of single cell omics data and new cellular models, there is increasing interest in the study of early human development, owing in particular to the importance of this period in the establishment of a healthy pregnancy. The EGA is an important transition stage in human embryonic development, and it is not adequately characterized at the molecular level. This study contributes to our knowledge of transcription factor networks that are likely to have importance at this stage of embryonic development. This type of proteomics analysis is significant because it lends experimental weight to speculation based on transcriptome or chromatin binding/accessibility studies. This analysis is thorough and comprehensive, and targets many transcription factors and chromatin modifiers that are likely to be functional in the EGA. Orthogonal strategies at several levels enhance the confidence in the biochemical data.

It is not until the end of the main section of the manuscript (under "Limitations")

that it becomes evident that all studies were performed in a variant of the HEK293 immortalized cell line. It is remarkable that so many of these developmental proteins could even be detected in this cell type. The strategy behind the use of 293 versus a more developmentally relevant model has to do with the practicalities of the experiments, but this strategy should be made clear up front. The impact of the study would be increased if at least a few key interactions could be validated in for example the 8C system model described by Mazid et al., or in naive human pluripotent stem cells, though this may be beyond the capabilities of the authors' laboratory. Functional validation by knockdown would also be informative. Novel biological insights are thus limited but the study represents an important resource for future work

Specific comments:

1. Page 5 indicate in the results what cell line is used for the analysis and why
2. Page 6 explain MAC tag here briefly
3. Page 6 briefly explain mass spec microscopy here.
4. Figure 2c a cutoff of 1 RKPM is not particularly stringent-this graph is not very informative
5. Figure 2d comment on enrichment of beta-catenin/TCF complex assembly? Wnt pathway appears again in GO analysis in Figure 7a
6. Page 6-what might account for enrichment of activation of Hox gene during differentiation
7. Figure 3c is the large proportion of novel interactions surprising if these can be detected in 293 cells?
8. Figure 7b how were these particular examples selected?

Referee #2:

This study concerns transcription factors expressed at the time of zygotic genome activation in the early human embryo (between 2-cell and 16-cell). Gawrisycki and colleagues performed mass spectrometry-based interaction analysis with tagged transcription factors to identify candidate interacting proteins. They then performed ChIPseq to localise binding sites for these factors taking advantage of the introduced tag. The authors claim to have identified >1000 interactions for the 21 transcription factors examined and to have "derived comprehensive regulatory networks". However, all studies were carried out in immortalised kidney cells (HEK293) with no validation, let alone perturbation, in early embryos or pluripotent stem cells. The results are therefore no more than a list of putative but unproven protein interactions. Leaving aside technical noise, the list is necessarily both incomplete and inaccurate (false or irrelevant positives) because it has not been generated in a relevant cell type. The data might be a resource for other investigations but as a standalone do not provide any mechanistic advance or

biological insight.

As a resource the content could be improved by including additional datasets in the human embryo RNAseq comparison, notably Petropoulos et al., 2016, using a higher RPKM cutoff and focussing in more detail on the 8-cell stage which is when major ZGA occurs.

The authors concluding statement that it would be challenging to perform studies in pluripotent stem cells is disingenuous as these cells can be expanded in very large numbers and have been used for many proteomics studies.

** As a service to authors, EMBO Press provides authors with the possibility to transfer a manuscript that one journal cannot offer to publish to another EMBO publication or the open access journal Life Science Alliance launched in partnership between EMBO Press, Rockefeller University Press and Cold Spring Harbor Laboratory Press. The full manuscript and if applicable, reviewers' reports, are automatically sent to the receiving journal to allow for fast handling and a prompt decision on your manuscript. For more details of this service, and to transfer your manuscript please click on

Link Unavailable

Dear Dr. Anderson,

We sincerely appreciate the time and effort expended by the reviewers and the editor in evaluating our manuscript entitled, "Interaction network of human early embryonic transcription factors", submitted to the EMBO Journal. We are grateful for the constructive feedback, which we believe has significantly helped in improving the quality and impact of our manuscript. We are writing to address the concerns raised by the reviewers and to provide a detailed response to each comment. We believe that we have adequately addressed all the reviewers' concerns and made necessary revisions to our manuscript, as outlined below.

Referee #1:

Gawriyski et al.

In this study Gawriyski et al. use a combination of structure prediction, proteomics methodology, CHIP-seq and computational analyses to define the transcription factor networks that are likely to operate during embryonic genome activation (EGA). The authors focused on the ETCHbox family because this group of TF had been previously implicated in EGA. Structural predictions identified DNA binding domains and transactivation domains for a number of family members. The authors then used affinity purification mass spectrometry and proximity dependent biotin identification to discover protein protein interactions for a subset of the genes of interest. Thereafter more interactions were identified using a wider set of developmentally relevant TF in the interactome analysis. Ontology and expression studies further supported the likely role of high confidence interactions identified in transcriptional regulation (transcription factors or chromatin modifiers) during early development. CHIP-seq studies revealed that many factors interacted with binding sites of genes active during the eight cell stage of development (a major wave of EGA).

With the availability of single cell omics data and new cellular models, there is increasing interest in the study of early human development, owing in particular to the importance of this period in the establishment of a healthy pregnancy. The EGA is an important transition stage in human embryonic development, and it is not adequately characterized at the molecular level. This study contributes to our knowledge of transcription factor networks that are likely to have importance at this stage of embryonic development. This type of proteomics analysis is significant because it lends experimental weight to speculation based on transcriptome or chromatin binding/accessibility studies. This analysis is thorough and comprehensive, and targets many transcription factors and chromatin modifiers that are likely to be functional in the EGA. Orthogonal strategies at several levels enhance the confidence in the biochemical data.

We thank the reviewer for appreciating our efforts, and statements such as "this type of proteomics analysis is significant because it lends experimental weight to speculation based on transcriptome or chromatin binding/accessibility studies" and "this analysis is thorough and comprehensive, and targets many transcription factors and chromatin modifiers that are likely to be functional in the EGA. Orthogonal strategies at several levels enhance the confidence in the biochemical data."

It is not until the end of the main section of the manuscript (under "Limitations") that it becomes evident that all studies were performed in a variant of the HEK293 immortalized cell line. It is remarkable that so many of these developmental proteins could even be detected in this cell type. The strategy behind the use of 293 versus a more developmentally relevant model has to do with the practicalities of the experiments, but this strategy should be made clear up front. The impact of the study would be increased if at least a few key interactions could be validated in for example the 8C system model described by Mazid et al., or in naive human pluripotent stem cells, though this may be beyond the capabilities of the authors' laboratory. Functional validation by knockdown would also be informative. Novel biological insights are thus limited but the study represents an important resource for future work

We appreciate the reviewer's concern regarding the choice of cell line used in our study. HEK293s nor any other cell line is not a 'perfect' model of preimplantation and we have expanded upon this in the 'Limitations of the Study' chapter. We checked that the cell line is clearly mentioned in every relevant chapter. Flp-In™ T-REx™ 293 cell line was used as it is the gold-standard in interaction proteomics with a large number of signaling pathways active and with very large range of genes expressed. Additionally, the inducible expression system it offers, without transgene expression leakage, is a must for studying key signaling molecules of which expression can transform the cells or be toxic to the. In our recent proteomics study of the embryonic factor DUX4 the model proved to be extremely useful (Vuoristo et al. 2022), as DUX4 expression is lethal to cells and will kill most iPSCs in less than 6 hours, which is not enough for inducing meaningful transgene levels in the cells.

However, in this revised manuscript we performed validation analysis with the BioID analysis in human induced pluripotent stem cell line (iPSC) HEL24.3. We could obtain three cell lines lines expressing MAC3-tagged DUXA, TPRX2 and ZSCAN4. Generation of the whole dataset with this model is not feasible, or in the scope of this manuscript, due to the challenges for obtaining cell lines expressing all the factors which we could analyse in the Flp-In™ T-REx™ 293 cell line. However, the interactome obtained from iPSC cell lines showed high overlap with the BioID-MS HCIs produced in the HEK293 cell line (as shown in Figure S11a). Notably, the DUXA interactome overlap was at a prominent level of 91.7% (Figure S11a). Similarly, TPRX2 and ZSCAN4 overlaps were 69.1% and 84.4%, respectively (Figure S11a). The difference in the overlap with TPRX2 most likely results from the better sensitivity of the HEK293 model, as all the HEK293 specific interactions are linked to regulation of transcription (Figure S11B).

Additionally, we performed PPI validation to assess the accuracy of the interaction information we provided with the Flp-In™ T-REx™ 293 cells. 96 protein interaction pairs (detected with >20 PSM values) were selected and validated via co-expression co-immunoprecipitation (co-IP) dotblot (Figure S12, Table S8). The validation ratio was 82% (79/96 of the tested interaction pairs validated). In sum, the results from both validation methods suggest that the interactions of the preimplantation interaction network can be reliably replicated and are biologically relevant.

Specific comments:

1. Page 5 indicate in the results what cell line is used for the analysis and why

We have expanded upon the choice of the cell line on page 5, as well as the limitations of the study chapter. Also see comment above.

2. Page 6 explain MAC tag here briefly

We have added a brief explanation of the MAC tag on page 5, along with references for further understanding.

3. Page 6 briefly explain mass spec microscopy here.

A brief explanation of Mass Spec Microscopy has been provided on page 6, with references for further reading.

4. Figure 2c a cutoff of 1 RKPM is not particularly stringent-this graph is not very informative

We expanded the comparative analysis with embryonic transcriptomic datasets to other equivalent datasets in addition to Yan et al. 2013 (Petropoulos et al. 2016, Zou et al. 2022) and 8-cell like cells (Mazid et al. 2022, Yoshihara et al. 2022, Yu et al. 2022, Taubenschmid-Stowers et al. 2022). For the comparison with embryonic cell transcriptomics studies or other comparisons with data extracted from databases, we used the cutoffs mentioned in the original papers mentioned either as limit of detection or limit of a gene being considered expressed in the original papers.

*Human Protein Atlas, NX ≥ 1 are considered detected.
(<https://www.proteinatlas.org/humanproteome/subcellular/cell+line>)*

*Yan et al. 2013, RPKM ≥ 1
Petropoulos et al. 2016, RPKM ≥ 1
Zou et al. 2022, FPKM > 1*

Additional figures added (Supplementary Figure 7, Supplementary Figure 8).

We are happy to provide the figures in other cut-offs if the reviewer wishes, however we deemed that the cut-offs set by the authors of the original articles for expression/detection would be reliable.

5. Figure 2d comment on enrichment of beta-catenin/TCF complex assembly? Wnt pathway appears again in GO analysis in Figure 7a

We have included a comment on the enrichment of the beta-catenin/TCF complex assembly and its relation to the Wnt pathway and the role of Wnt/ β -catenin involvement in gene regulation during embryonic development in the text accompanying Figure 2d.

6. Page 6-what might account for enrichment of activation of Hox gene during differentiation

We have now included a discussion regarding the potential reasons behind the enrichment of Hox gene activation during differentiation.

7. Figure 3c is the large proportion of novel interactions surprising if these can be detected in 293 cells?

The prior information in the proteomics databases were from hybrid arrays (e.g., yeast two hybrid or similar). No systematic analyses for these proteins existed before our study. The results presented in this paper are the first interaction proteomics (AP-MS and BioID-MS) results for these proteins (excluding the LEUTX interactome which we published earlier). The used methods represent the cutting-edge gold-standard on defining protein-protein interactions. A larger number of interactions was expected in most cases. The number of interactions detected in this manuscript are in agreement on what we recently published for >100 human transcription factors (Göös et al., 2022, Nature Communications).

8. Figure 7b how were these particular examples selected?

We have provided the criteria used for selecting the examples shown in Figure 7b in the accompanying text. The key criteria were regions that peak in expression in embryonic stages (4- or 8-cell of Wu et al. 2018 ATAC-Seq data) and are lower in primed hESC stages in particular. The genes chosen are only a subset. In the case of CRX, ZSCAN4 and DUXA they are highlighted also because they are baits in the proteomics results.

Referee #2:

This study concerns transcription factors expressed at the time of zygotic genome activation in the early human embryo (between 2-cell and 16-cell). Gawrisycki and colleagues performed mass spectrometry-based interaction analysis with tagged transcription factors to identify candidate interacting proteins. They then performed ChIPseq to localise binding sites for these factors taking advantage of the introduced tag. The authors claim to have identified >1000 interactions for the 21 transcription factors examined and to have "derived comprehensive regulatory networks". However, all studies were carried out in immortalised kidney cells (HEK293) with no validation, let alone perturbation, in early embryos or pluripotent stem cells. The results are therefore no more than a list of putative but unproven protein interactions. Leaving aside technical noise, the list is necessarily both incomplete and inaccurate (false or irrelevant positives) because it has not been generated in a relevant cell type. The data might be a resource for other investigations but as a standalone do not provide any mechanistic advance or biological insight.

As a resource the content could be improved by including additional datasets in the human embryo RNAseq comparison, notably Petropoulos et al., 2016, using a higher RPKM cutoff and focussing in more detail on the 8-cell stage which is when major ZGA occurs.

The authors concluding statement that it would be challenging to perform studies in pluripotent stem cells is disingenuous as these cells can be expanded in very large numbers and have been used for many proteomics studies.

We acknowledge the reviewer's concern regarding validation in a more developmentally relevant model and validation of protein-protein interactions through another applicable method. We also agree that HEK293s nor any other cell line is not a 'perfect' model of preimplantation and we have expanded upon this in the 'Limitations of the Study' chapter. Flp-In™ T-REx™ 293 cell line was used as it is the gold-standard in interaction proteomics with a large number of signaling pathways active and with large number and very wide range of genes expressed. Additionally, the inducible expression system it offers, without transgene expression leakage, is a must for studying key signaling molecules of which expression can transform the cells or be toxic to the. In our recent proteomics study of the embryonic factor DUX4 the model proved to be extremely useful (Vuoristo et al. 2022), as DUX4 expression is lethal to cells and will kill most iPSCs in less than 6 hours, which is not enough for inducing meaningful transgene levels in the cells.

However, in this revised manuscript we performed validation analysis with the BioID analysis in human induced pluripotent stem cell line (iPSC) HEL24.3. We could obtain three cell lines expressing MAC3-tagged DUXA, TPRX2 and ZSCAN4. Generation of the whole dataset with this model is not feasible, or in the scope of this manuscript, due to the challenges for obtaining cell lines expressing all the factors which we could analyse in the Flp-In™ T-REx™ 293 cell line. However, the interactome obtained from iPSC cell lines showed high overlap with the BioID-MS HCIs produced in the HEK293 cell line (as shown in Figure S11a). Notably, the DUXA interactome overlap was at a prominent level of 91.7% (Figure S11a). Similarly, TPRX2 and ZSCAN4 overlaps were 69.1% and 84.4%, respectively (Figure S11a). The difference in the overlap with TPRX2 most likely results from the

better sensitivity of the HEK293 model, as all the HEK293 specific interactions are linked to regulation of transcription (Figure S11B).

Additionally, we performed PPI validation to assess the accuracy of the interaction information we provided with the Flp-In™ T-REx™ 293 cells. 96 protein interaction pairs (detected with >20 PSM values) were selected and validated via co-expression co-immunoprecipitation (co-IP) dotblot (Figure S12, Table S8). The validation ratio was 82% (79/96 of the tested interaction pairs validated). In sum, the results from both validation methods suggest that the interactions of the preimplantation interaction network can be reliably replicated and are biologically relevant.

Based on the reviewer's suggestion on human embryo RNA-Seq comparison, we have included additional embryonic transcriptomic datasets from the mentioned references (Petropoulos et al. 2016) as well new reference (Zou et al. 2022) and refocused our analysis on the 8-cell stage, where major ZGA occurs. Further, we have added comparative analysis with the 8-cell like cell datasets Mazid et al. 2022, Yoshihara et al. 2022, Taubenschmid-Stowers et al. 2022 and Yu et al. 2022. The manuscript has been updated to reflect these changes. The cutoffs used in these analyses are the ones provided as cutoffs for detection or expression in the original papers.

Date: 13th Dec 23 10:06:32

Last Sent: 13th Dec 23 10:06:32

From: k.anderson@embojournal.org

To: markku.varjosalo@helsinki.fi

Subject: EMBOJ-2023-113907R-Q Decision Letter

Message: Dear Prof. Varjosalo,

Thank you for submitting your manuscript (EMBOJ-2023-113907R-Q) to The EMBO Journal. I have now had a chance to read it carefully and to discuss it with my colleagues, and I am sorry to say that we cannot offer publication in The EMBO Journal. As you will see, the referees agree there are improvements in the revision, however their internal rankings and reports unfortunately indicate the data are more well-suited to a specialist audience rather than a more general readership.

That said, we still found this work suitable for our sister journal EMBO reports, in light of their focus on interesting key observations that do not necessarily need to be fully mechanistically followed up. I therefore discussed the work with my EMBO reports colleague, Dr. Esther Schnapp, who considered the study interesting and would be happy to work with the existing reports to move forward in case you transfer it to EMBO reports. Should you be interested in this option, please simply follow the transfer link; no reformatting is required.

Yours sincerely,

Kelly M Anderson, PhD
Editor, The EMBO Journal
k.anderson@embojournal.org

Referee #1:

In their revised manuscript, the authors address all concerns raised in my previous review in a thorough and satisfactory fashion. The revision makes the limitations of the experimental system clear, but adds new data from a more relevant cell line, and addresses a number of minor points of clarity. The data in this study will be very useful to those interested in early human development and in vitro embryo models.

Referee #2:

In this revised manuscript Gawrisyki and colleagues have added further analyses of embryo expression, detected interactions for 3 factors in human pluripotent stem cells, and performed a co-immunoprecipitation validation assay for 96 of the protein interaction pairs. Overall these additions support the reliability of their data and their potential utility as a resource. The paper still lacks any functional experiments which one would normally expect in an EMBO J paper, however. Regarding the interaction study in human pluripotent stem cells, the authors reasonably highlight the overlap with interactions found in HEK293 cells. They do not comment, however, on whether they found any additional interactions and the presentation of data in Fig S11a does not reveal this. It would be interesting to comment on any additional interactions if they are present. A minor comment is that the authors comment on the role of Wnt/bcatenin in embryonic stem cells. This is not relevant to ZGA.

** As a service to authors, EMBO Press provides authors with the possibility to transfer a manuscript that one journal cannot offer to publish to another EMBO publication or the open access journal Life Science Alliance launched in partnership between EMBO Press, Rockefeller University Press and Cold Spring Harbor Laboratory Press. The full manuscript and if applicable, reviewers' reports, are automatically sent to the receiving journal to allow for fast handling and a prompt decision on your manuscript. For more details of this service, and to transfer your manuscript please click on *Link Unavailable*

Dear Markku,

Thank you for the transfer of your revised manuscript to EMBO reports. We can in principle accept it, only a few more minor revisions will be required. Also, please address the last comments by referee 2 and please provide a detailed point-by-point response to all final requests with your final ms.

- Please rename the conflict of interest subheading to "Disclosure and Competing Interest Statement"
- Please correct the discrepancy between Zenglai Tan in the ms file vs. Tan Zenglai in our online submission system.
- Please remove the author credits from the ms file. All credits need to be entered online during ms submission.
- The reference format is not correct, please correct to the EMBO reports style. The section heading should be References, et al needs to be used after 10 author names, and DOIs should only be used for preprints and datasets that have not been published yet.
- Please send us a completed author checklist that can be found here: <https://www.embopress.org/page/journal/14693178/authorguide>. Please note that the completed checklist will also be part of our transparent peer-review process file.
- Please add all funding info also in our online ms submission system.
- Please add the missing callouts for Figures 2a, 6e; Figure 10b is called out before Figure 6a, please correct.
- There are 9 uploaded datasets but the nomenclature and callouts are incorrect - they need to be called Dataset EV1, etc. instead of Supplemental Table 1 etc.; their legends need to be removed from the ms file and need to be provided in each Excel file.
- The Abbreviations section needs to be removed from the manuscript. Abbreviations should be defined in brackets after their first mention in the text, not in a list of abbreviations.
- The manuscript sections should be in the following order: Title page - Abstract & Keywords - Introduction - Results - Discussion - Materials & Methods - Data Availability - Acknowledgments - Disclosure Statement & Competing Interests - References - Figure Legends - Tables with legends - Expanded View Figure Legends.
- There are 13 separately uplidd Supplemental figures whose legends are in the ms file; if some of them will be in the manuscript then their legends can stay but the nomenclature should then be Figure EV1, etc. the rest of the figures can be placed in an Appendix file with nomenclature Appendix Figure S1, etc. You can find more info on our supplementary file types in our guide to authors online: <https://www.embopress.org/page/journal/14693178/authorguide#expandedview>
- The Lead contact section should be removed from the ms.
- Please note that the specific URLs for PRJEB58808 and MSV000091321 datasets need to be provided in the data availability statement.
- Please note that reviewer access codes for PRJEB58808 and MSV000091321 datasets need to be provided in the data availability statement.
- Figure legends: Please specify the statistical test used for data analysis in the legends of figures 5c; 6d; 7a.
- Please write the abstract in present tense, as per journal policy.

EMBO press papers are accompanied online by A) a short (1-2 sentences) summary of the findings and their significance, B) 2-3 bullet points highlighting key results and C) a synopsis image that is exactly 550 pixels wide and 200-600 pixels high (the height is variable). You can either show a model or key data in the synopsis image. Please note that text needs to be readable at the final size. Please send us this information along with the final manuscript.

Referee 1:

In their revised manuscript, the authors address all concerns raised in my previous review in a thorough and satisfactory fashion. The revision makes the limitations of the experimental system clear, but adds new data from a more relevant cell line, and addresses a number of minor points of clarity. The data in this study will be very useful to those interested in early human development and in vitro embryo models.

Referee 2:

In this revised manuscript Gawrisycki and colleagues have added further analyses of embryo expression, detected interactions for 3 factors in human pluripotent stem cells, and performed a co-immunoprecipitation validation assay for 96 of the protein interaction pairs. Overall these additions support the reliability of their data and their potential utility as a resource. The paper still lacks any functional experiments which one would normally expect in an EMBO J paper, however. Regarding the interaction study in human pluripotent stem cells, the authors reasonably highlight the overlap with interactions found in HEK293 cells. They do not comment, however, on whether they found any additional interactions and the presentation of data in Fig S11a does not reveal this. It would be interesting to comment on any additional interactions if they are present. A minor comment is that the authors comment on the role of Wnt/bcatenin in embryonic stem cells. This is not relevant to ZGA.

Referee 2:

In this revised manuscript Gawrisycki and colleagues have added further analyses of embryo expression, detected interactions for 3 factors in human pluripotent stem cells, and performed a co-immunoprecipitation validation assay for 96 of the protein interaction pairs. Overall these additions support the reliability of their data and their potential utility as a resource. The paper still lacks any functional experiments which one would normally expect in an EMBO J paper, however.

Regarding the interaction study in human pluripotent stem cells, the authors reasonably highlight the overlap with interactions found in HEK293 cells. They do not comment, however, on whether they found any additional interactions and the presentation of data in Fig S11a does not reveal this. It would be interesting to comment on any additional interactions if they are present.

A minor comment is that the authors comment on the role of Wnt/bcatenin in embryonic stem cells. This is not relevant to ZGA.

We have now removed the comment on the role of Wnt/bcatenin in embryonic stem cells.

Editorial requests:

Please rename the conflict of interest subheading to "Disclosure and Competing Interest Statement"

done

The section heading should be References

done

Please add the missing callouts for Figures 2a, 6e; Figure 10b is called out before Figure 6a, please correct.

done

The Abbreviations section needs to be removed from the manuscript. Abbreviations should be defined in brackets after their first mention in the text, not in a list of abbreviations.

done

The Lead contact section should be removed from the ms.

done

Figure legends: Please specify the statistical test used for data analysis in the legends of figures 5c; 6d; 7a.

done

Please note that the specific URLs for MSV000091321 datasets need to be provided in the data availability statement.

done

The manuscript sections should be in the following order: Title page - Abstract & Keywords - Introduction - Results - Discussion - Materials & Methods - Data Availability - Acknowledgments - Disclosure Statement & Competing Interests - References - Figure Legends - Tables with legends - Expanded View Figure Legends.

done

Please write the abstract in present tense, as per journal policy.

done

Please remove the author credits from the ms file. All credits need to be entered online during ms submission.

done

There are 9 uploaded datasets but the nomenclature and callouts are incorrect - they need to be called Dataset EV1, etc. instead of Supplemental Table 1 etc.; their legends need to be removed from the ms file and need to be provided in each Excel file.

done

- Please send us a completed author checklist that can be found here:<<https://www.embopress.org/page/journal/14693178/authorguide>>;. Please note that the completed checklist will also be part of our transparent peer-review process file.

done

- Please note that reviewer access codes for PRJEB58808 and MSV000091321 datasets need to be provided in the data availability statement.

Direct links added, now access codes needed

- Please correct the discrepancy between Zenglai Tan in the ms file vs. Tan Zenglai in our online submission system.

Done

- Please add all funding info also in our online ms submission system.

done

- There are 13 separately uplidd Supplemental figures whose legends are in the ms file; if some of them will be in the manuscript then their legends can stay but the nomenclature should then be Figure EV1, etc. the rest of the figures can be placed in an Appendix file with nomenclature Appendix Figure S1, etc. You can find more info on our supplementary file types in our guide to authors online: <https://www.embopress.org/page/journal/14693178/authorguide#expandedview>

done

- The reference format is not correct, please correct to the EMBO reports style. et al needs to be used after 10 author names, and DOIs should only be used for preprints and datasets that have not been published yet.

done

Prof. Markku Varjosalo
University of Helsinki
Institute of Biotechnology
Biocenter 3, P.O.Box 65 (Viikinkaari 1)
Helsinki, orcid||||| 00014 UH
Finland

Dear Prof. Varjosalo,

I am very pleased to accept your manuscript for publication in the next available issue of EMBO reports. Thank you for your contribution to our journal.

Yours sincerely,
